# Max Explainability Score with Confidence Interval (MES-CI): A Quantitative Metric for Interpretability in Knowledge Graph-Based Recommender Systems

## Abstract

Knowledge graph-based recommender systems (KGRS) utilize structured semantic relationships to generate personalized and interpretable recommendations, leveraging the inherent connectivity within knowledge graphs to enhance transparency. While KGRS offer significant advantages in explainability, quantifying the reliability and impact of these explanations remains challenging due to the complexity of underlying models and the multiple pathways that influence recommendation outcomes. This paper critically analyzes existing evaluation metrics for explainability in KGRS, identifying their limitations and advocating for a balanced framework that integrates interpretability with predictive accuracy. This research builds upon the existing Max Explainability Score (MES) by introducing an enhanced scoring mechanism, the Max Explainability Score with Confidence Interval (MES-CI). MES-CI overcomes the limitations of evaluating the explainability of generated recommendations using a single-point score by providing a more comprehensive and balanced assessment. It incorporates confidence intervals alongside confidence score percentages, offering a clearer representation of explainability reliability. Furthermore, the applicability of this refined metric is examined across multiple datasets, with case studies demonstrating its effectiveness in improving transparency and user trust in AI-driven recommendation systems.

## 1 Introduction

Recommender systems (RSs) are designed to suggest relevant items to users based on their preferences and can be developed using traditional approaches such as content-based filtering, collaborative filtering, and hybrid methods [1]. While these systems achieve high accuracy, they often lack transparency and fail to provide clear explanations for their recommendations. To address this limitation, explainable recommendation systems (XRS) have emerged, offering personalised recommendation algorithms specifically designed to clarify the reasoning behind suggested items. By incorporating detailed explanations, XRS enhances user trust and understanding, ensuring that recommendations are not only accurate but also interpretable [2].

XRS are predominantly built using knowledge graphs (KG), which are structured as *subject–predicate–object* (SPO) triples within a heterogeneous graph. The KGs have gained significant traction for their ability to harness structured semantic relationships, enabling more precise and personal recommendations. By enhancing contextual understanding, KGs address limitations present in traditional recommender models, offering deeper insights into user preferences [3]. However, while KGs contribute to explainability, quantifying or systematically evaluating this interpretability remains a challenge [4].

Submitted to 39th Conference on Neural Information Processing Systems (NeurIPS 2025). Do not distribute.

Researchers typically assess the effectiveness of XRS explainability through user studies, online experiments, and offline evaluations based on historical datasets. For instance, Cao et al. [5] assessed the effectiveness of their PR4SR model by conducting a study with 50 participants. Each participant was assigned 30 randomly selected cases from the Beauty, Cellphones, and Baby datasets. The study included a detailed questionnaire that provided information such as the user's session history, the starting node for path reasoning, and an explanation of the generated path. Participants were asked to evaluate the selection of the starting node and the explainable paths based on their comprehension of the session history. Similarly, Liu et. al. [6] conducted a crowdsourced evaluation, enlisting the 100 most active users to assess the explanations provided by their Ante-RNN model. However, explainability evaluation metrics derived from qualitative user studies often remain subjective and prone to confirmation biases [7], posing challenges in obtaining objective assessments of interpretability.

While Rosenfeld [8] introduced a quantitative explainability metric, it still requires human interaction to evaluate discrepancies between the agent model and its corresponding logical explanation. Tiwary et al. [9] proposed the Max Explainability Score (MES) as a quantitative metric for assessing recommendation explainability within KG-based frameworks. MES quantifies explainability by capturing surprising information, which is influenced by factors such as attributable features, their quality (i.e., the probability of occurrence in the recommendation generation path), information value (i.e., entropy derived from traversal paths), the relevance of the recommended item, and the rewards associated with traversal paths. However, MES relies on a single-point evaluation. A more comprehensive approach would incorporate confidence interval limits alongside the evaluation score, such as a 95% confidence level, providing a more detailed and reliable assessment of explainability.

This paper seeks to address the gap in explainability evaluation for knowledge graph-based recommender systems (KGRS) by reviewing existing metrics and enhancing the MES. Specifically, we extend MES beyond a single-point score by incorporating a confidence interval, offering a more reliable assessment of explainability. Through case studies, we validate the proposed explainability metric's applicability across diverse domains, setting a foundation for future research in XRS.

The primary contributions of this study encompass the following key aspects:

- A comprehensive review of current quantitative explainability metrics for KGRS.
- An extension of MES to include confidence interval-based scoring, ensuring a more nuanced evaluation of explainability.

This paper is structured as follows: Section 2 reviews background and related work on XRS explainability. Section 3 outlines the proposed explainability measurement algorithm, while Section 4 details the experimental framework. Section 5 provides an in-depth analysis, including a case study on evaluation mechanisms. Finally, Section 6 presents conclusions and future research directions.

## 2 Background and related work

This section outlines the foundational concepts and examines relevant research.

### 2.1 Why Explainability?

Explainability in AI refers to the ability to clearly convey the reasoning behind a model's output. It addresses critical questions such as: How does the system operate? What factors influence specific predictions? Why were particular recommendations made for the user? Which explanation best supports a given prediction or recommendation? By providing transparency into data processing mechanisms, explainability not only enhances user comprehension but also helps uncover potential biases and limitations within the system. A well-explained AI model fosters trust and confidence, making it more reliable for both users and decision-makers. Since many AI models function as black boxes, it is crucial to ensure users have clear insights into how predictions or recommendations are generated [10]. According to GDPR and other relevant regulations, users have the fundamental right to understand the rationale behind AI-generated decisions. Transparency in AI fosters greater trust and engagement, reinforcing its credibility and encouraging wider adoption in real-world applications.

Recent advancements in KGs, reinforcement learning (RL), and large language models (LLMs) are playing a significant role in shaping explainable AI (XAI), enabling AI systems to provide clearer, more interpretable insights [11]. As RSs become increasingly sophisticated, ensuring the validity and

reliability of their generated explanations is crucial. Without proper validation, explanations may lack consistency or fail to provide meaningful insights into how recommendations are derived. Therefore, there is a growing need for robust evaluation frameworks that assess the quality, relevance, and impact of explanations, ultimately improving transparency and accountability in AI-driven decision-making.

## 2.2 Explainability implementation in XRS

Explainability in AI is achieved through two main approaches: embedded methodologies and post-hoc techniques. Embedded methods integrate explainability directly within the AI model, ensuring that the reasoning behind predictions is generated internally before producing an outcome. In contrast, post-hoc techniques analyze decisions after the output has been generated, using methods such as feature importance analysis, surrogate models, and counterfactual reasoning to provide insights into the model's decision-making process [3].

To enhance XRS, researchers incorporate KGs into RSs as auxiliary information, improving both performance and interpretability. This integration leverages structured relationships between entities, making recommendations more intuitive and explainable. Several KG-based RS methodologies have been explored, including post-hoc approaches, KGE methods, path-based embedding strategies, and unified frameworks, each contributing to improved explainability and user trust in AI-generated recommendations [2].

Post-hoc explainability methods generate explanations after model development by applying soft matching algorithms to enhance recommendation transparency. A widely recognized post-hoc approach for assessing model outcomes is determining the significance of various features in influencing predictions. Feature importance can be quantified using post-hoc methods, such as Shapley values, which are derived from game theory principles [12]. This method enables recommendations to align more closely with user preferences while ensuring interpretability.

KGE-based approaches integrate explainability directly into the recommendation generation process by embedding structured semantic relationships. These methods determine entity similarity by evaluating the distance between their embeddings, often integrating item-side attributes [13] or user preferences [14] within user-item KGs. By embedding entities effectively, these methods enhance both prediction accuracy and explanation consistency.

Path-based recommendation techniques leverage the KG structure and its meta-paths to infer meaningful connections between users and items. Li et al. [15] proposed the complex-to-concise (C2C) meta-multigraph, designed to facilitate message propagation from complex structures to more concise representations as it traverses the depth of the meta-multigraph. Xian et al. [16] developed a policy-guided path reasoning algorithm that applies reinforcement learning (RL) to identify the most relevant paths between user-item pairs, optimizing recommendation reliability while maintaining explainability.

Unified approaches combine embedding-based and path-based techniques, benefiting from the strengths of both semantic embeddings and structured KG traversal patterns. These hybrid methods aim to maximize interpretability while retaining high predictive accuracy, ensuring that recommendations are both transparent and relevant. By integrating entity representations with structured reasoning frameworks, unified methods improve user trust and facilitate the deployment of AI-driven personalized RSs [3].

Overall, leveraging KGs in RSs enhances explainability by providing structured, interpretable relationships between entities, allowing AI systems to generate recommendations that align with user expectations while maintaining transparency. As research progresses, developing more robust frameworks that balance interpretability, and predictive performance will remain crucial for advancing explainable AI-driven recommendation technologies.

## 2.3 Explainability evaluation in XRS

XRS algorithms are traditionally evaluated using standard metrics such as *root mean square error (RMSE)* and *mean absolute error (MAE)* for rating predictions, as well as *accuracy*, *recall*, *Hit Rate*, *F-measure*, and *normalized discounted cumulative gain (NDCG)* for top-n recommendations. While these metrics provide insight into the predictive performance of RSs, assessing explainability requires additional methodologies [17].

### 2.3.1 Explainability - qualitative evaluation

Qualitative evaluation of explainability in RS increasingly incorporates user-centered methods such as user studies, offline simulations, and online A/B testing to assess user satisfaction, trust, and perceived transparency. User studies involve direct interaction with participants through interviews, surveys, or usability tasks to gather subjective feedback on the quality and helpfulness of explanations. Offline assessments use historical data to simulate user responses and analyze interpretability or transparency metrics. Meanwhile, online assessments track real-time user behavior, such as dwell time, conversion rates, and interaction patterns, to measure the actual impact of explanations on user engagement and decision-making [18]. These approaches, although effective, are constrained by practical limitations such as sample size, scalability, and the subjective nature of human feedback.

### 2.3.2 Explainability - quantitative evaluation

Quantitative evaluation of explainability in AI systems is demonstrated in prior work [8], where explainability measures are derived from performance variations across models of different fidelity. These measures include the number of rules in the generated explanations, the quantity of features utilized for explanation generation, and the stability of the system's explanatory framework. Despite its contributions, this approach does not explicitly address explainability evaluation in RSs, particularly those leveraging KGs and related technologies. Additionally, it requires human interaction to compare the system's model outputs with the logical reasoning presented in its explanations, adding another layer of complexity.

Recently, Tiwary et al. [9], proposed MES, a quantitative metric designed to evaluate the explainability of AI-driven RSs, particularly those leveraging KG. Unlike traditional explainability measures that rely on subjective user studies or post-hoc evaluations, MES provides a structured approach to assessing how well a RS conveys transparent and interpretable insights. MES quantifies explainability by capturing surprising information, which is influenced by several key factors:

- *Attributable Features*: The characteristics contributing to a recommendation, ensuring that users can trace the reasoning behind suggested items.

- *Feature Quality*: The probability of occurrence of these features within the recommendation generation path, reflecting their reliability.

- *Information Value*: The entropy derived from traversal paths in the KG, indicating how much new or unexpected information is introduced.

- *Rewards Assigned to Traversal Paths*: The significance of different paths taken within the KG to arrive at a recommendation, reinforcing the credibility of the explanation.

MES is particularly valuable in KGRS, where structured semantic relationships can enhance transparency. By advancing beyond conventional explainability scores, MES contributes to the development of fair, user-centric, and accountable AI-driven RSs, reinforcing trust and usability in AI applications.

### 2.4 Analysis of MES

The MES formula is structured around two key components of explainability: Reward gain and Information gain. The central question is whether these parameters are meaningful within the context of XAI and KGRS? AI models generate predictions and recommendations by analyzing structured relationships within data, ensuring that outputs align with historical user behaviors, actions, and future needs. This interconnected approach allows AI systems to uncover patterns in user interactions, providing insights into preferences and decision-making. [19]. KGs play a crucial role in enhancing explainability by systematically capturing and representing entity relationships. They enable AI models to traverse structured pathways, offering a clear rationale behind recommendations. This graph-based traversal serves two essential purposes: selecting relevant items for users and justifying those selections with interpretable reasoning. Unlike opaque black-box models, KG-based AI systems provide transparency, fostering trust and confidence [20]. However, a key challenge lies in determining the most reliable explanatory pathway among multiple connections between users and recommended items. Not all connections hold equal importance, and identifying the most interpretable and relevant explanation requires well-defined methodologies.

This is where the MES framework introduces reward gain and entropy gain to quantify explainability.

- *Reward gain*: Reflects the value or significance of a traversal path in the KG. The model assigns rewards to specific paths based on their relevance, credibility, and alignment with user preferences. The higher the reward for a path, the more logically grounded and interpretable the explanation.

- *Information gain*: Captures the information richness and surprise factor of a given path. A recommendation is more explainable if the traversal path provides meaningful new information rather than redundant or overly predictable connections.

The explainability of a recommendation is influenced by the reward assigned to a path within the KG and the quality of information it provides. A well-designed explainability framework must strike a balance between semantic relevance (ensuring recommendations are logically grounded) and novel insight (offering valuable, non-trivial information). This equilibrium fosters a deeper understanding, trust, and engagement among users interacting with AI-generated recommendations, ultimately driving broader adoption and improved usability of interpretable AI systems.

A key limitation of MES in its original form is its reliance on single-point evaluation, which may not fully capture the variability and confidence in explainability assessments. To address this, we introduce confidence interval-based scoring, allowing for a more nuanced and equitable evaluation. By incorporating confidence scores—such as a 95% confidence level—the proposed metric offers a more robust measure of explainability, ensuring that recommendations are both interpretable and statistically sound. In the following section, we introduce our proposed approach, which quantitatively assesses the explainability performance of KGRS by integrating structured methodologies that balance interpretability and predictive accuracy.

# 3 Overviews of the proposed approach

The proposed approach aims to enhance the explainability evaluation of KGRS by introducing a refined framework built upon the MES. While traditional MES operates on single-point evaluation, the enhanced framework integrates confidence interval-based scoring, ensuring a more nuanced and statistically robust assessment of explainability. By establishing confidence scores, such as a 95% confidence level, we improve reliability and precision in measuring the transparency of recommendations.

The proposed approach, as shown in **Algorithm 1**, systematically assesses the *explainability score* of each path that connects a recommended product to a user, ensuring transparency and interpretability in AI-driven recommendations. The process begins by computing the *standard deviation* of all explainability scores associated with the product's various pathways. This step quantifies the variability in explainability across different paths, providing a measure of consistency and deviation within the recommendation framework.

Once the standard deviation is established, the approach proceeds to evaluate the *confidence interval*, specifically determining the 95% lower and upper bounds of the explainability score. This interval estimation enhances the robustness of the explainability assessment, offering a statistical range that accounts for potential fluctuations in how recommendations are justified.

Following this, the system identifies the path with the highest explainability score, selecting the most interpretable and relevant route that best aligns with user understanding and trust. The final output presents the *maximum explainability score (MES)* alongside its associated 95% confidence interval bounds, ensuring that the explanation is not only optimal but also statistically validated. By incorporating confidence intervals, this refined methodology strengthens the credibility of explainable recommendations, fostering greater user engagement and trust in AI-driven decision-making processes.

---

**Algorithm 1** Max Explainability Score with 95% Confidence Interval ($MES - CI$)

---

**Require:** Existing *Max Explainability Score (MES)* function
**Ensure:** Select the highest explainability score of the recommended product for the user.

   **for** $Prodcut_i \leftarrow$ Recommended List of Candidate $Products$ for $User$ **do**
      $ExplainabilityScore \leftarrow []$
      **for** $Path_i \leftarrow$ Paths Associated with $User$ and $Product_i$ **do**
         $ExpScore \leftarrow getExplainabilityScore$ of the associated $Path_i$
         $ExplainabilityScore \leftarrow$ Append $ExpScore$
      **end for**
      $STDExplainabilityScore \leftarrow$ Standatd Deviation of $ExplainabilityScore$
      **for** $Path_i \leftarrow$ Paths Associated with $User$ and $Product_i$ **do**
         $ExpScore \leftarrow getExplainabilityScore$ of the associated $Path_i$
         $ExpScore - CI_{Lower} \leftarrow ExpScore$ - $t_score$ ($ConfidenceScore$, dof = $len(ExplainabilityScore)$ -1 ) * ($STDExplainabilityScore$ / ($SQRT(len(ExplainabilityScore))$)))
         $ExpScore - CI_{Upper} \leftarrow ExpScore$ + $t_score$ ($ConfidenceScore$, dof = $len(ExplainabilityScore)$ -1 ) * ($STDExplainabilityScore$ / ($SQRT(len(ExplainabilityScore))$)))
      **end for**
      $MES \leftarrow MAX(ExplainabilityScore)$
      $MES - CI_{Lower} \leftarrow ExpScore - CI_{Lower}$ of $MES$
      $MES - CI_{Upper} \leftarrow ExpScore - CI_{Upper}$ of $MES$
      $MES - CI \leftarrow [MES, MES - CI_{Lower}, MES - CI_{Upper}]$
   **end for**
   **return** $MES - CI$

---

# 4 Experimentations

## 4.1 Dataset

The experiment utilizes domain-specific datasets from Amazon, focusing on two e-commerce categories: clothing and beauty, which were previously employed in prior research [9]. Each dataset comprises six key entities—user, product, product's feature words, related products, brand, and category—interconnected through eight distinct relationship types, as depicted in Figure 1. Users can purchase multiple products, each associated with specific categories and brands, while also exhibiting additional behavioral connections such as joint purchases, product views, and relationships with related products. Product descriptions incorporate feature words mentioned by users, further enriching the dataset. Together, these elements form user-centric KGs with clearly identifiable entities and structured relationships, facilitating more interpretable and personalized recommendations.

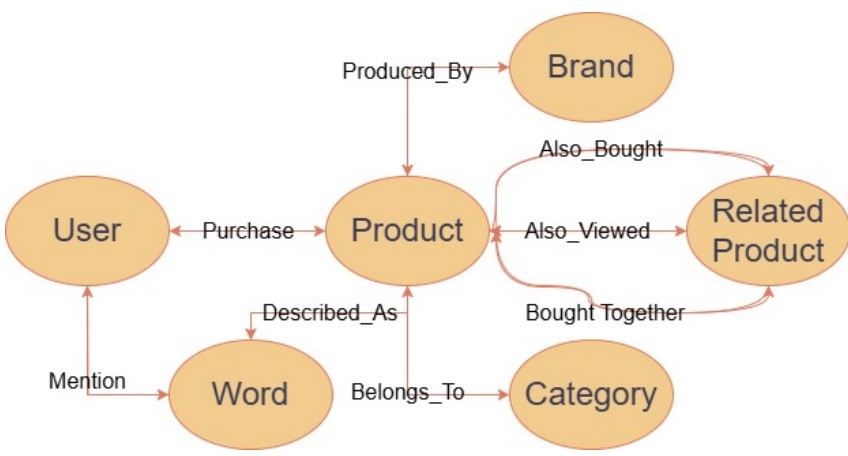

Figure 1: Amazon data - used in the experiment

## 4.2 Experimental methodology

Building on the previously established MES explainability framework, we have enhanced its capabilities while maintaining a similar experimental setup. The experiment involves developing an XRS model using RL based on users' historical purchases, with separate training (70%) and test (30%) datasets derived from transaction records. Initially, relevant entities—including product relationships and user reviews—are extracted from an application-specific e-commerce dataset. A label property graph (LPG) is employed for KG construction, structuring entities and relationships. The experiment then progresses to KGE generation using the TransE algorithm to create vector representations. The RL framework integrates KGE within an Actor-Critic model, where path traversal actions that align with predefined patterns are rewarded, while deviations incur penalties. Model effectiveness is evaluated using the test dataset, applying metrics such as NDCG, recall, hit rate (HR), and precision to assess recommendation quality. Ultimately, the explainability framework generates a quantitative explainability score with confidence interval, offering valuable insights into recommendation transparency and reinforcing interpretability in AI-driven decision-making.

## 5 Results and Analysis

This section presents an analysis of the experiments conducted with the proposed algorithm. Additionally, it explores the mechanisms used for explainability evaluation, demonstrated through a case study. Lastly, it examines the limitations inherent in the algorithm.

### 5.1 Validation sets and benchmarks

Having expanded the capabilities of the previously published explainability evaluation mechanism, MES, we adopted a similar evaluation framework as described in that paper. Both e-commerce datasets include corresponding test datasets, accounting for approximately 30% of the total transactions. To validate the model's performance, we utilized metrics such as NDCG, precision, HR, and recall.

The central focus of this paper is to evaluate the explainability of recommended items. Despite an extensive review of existing literature, we found MES to be the only reliable explainability evaluation mechanism, as it measures explainability by considering both information gain and reward gain. Building upon this foundation, we extend the metric by incorporating confidence intervals for the defined explainability measures, enhancing robustness and reliability. Consequently, we deliberately excluded discussions on model efficacy, limiting the scope strictly to the explainability of recommendations.

### 5.2 Explainability Evaluation - A Case Study

To assess explainability, consider an example from the Beauty dataset. User 21001 had previously purchased products [11808, 7381, 9141, 10747], as shown in Table 1. Based on these past purchases, the RS identifies path patterns to suggest new products. In this case, our KG and RL-based XRS generated a top-10 recommendation list: [11772, 8471, 9576, 5351, 1690, 5603, 8015, 1465, 10872, 5934]. The test dataset contains the actual products the user is expected to purchase in the future, which in this example is [11772]. This setup enables an evaluation of the system's predictive accuracy while offering insights into the transparency and interpretability of its recommendations.

Providing users with explanations for the recommended top-10 products would enhance their understanding of the model's decision-making process. Users may wonder how the system determines these recommendations, how it recognizes their preferences, and how it interprets their needs. Clear and well-structured explainability fosters trust in the RS, ensuring users feel confident in its relevance and utility.

This paper leverages the existing MES framework to enhance the explainability of the recommended top-10 products. As shown in Table 2, six possible explanations exist for recommending product 11772 to user 21001. However, determining which of these explanations is most meaningful and aligns with user preferences is essential. A key consideration is whether the user would find any of these explanations satisfactory or if they have specific criteria for evaluating them.

According to the MES framework, users seek to maximize both the rewards gained from the provided explanation and the informational value it conveys. In this scenario, among the six possible explana-

Table 1: Beauty dataset - A case study for user 21001 - Product recommendations

| User | Historical Purchases (train) | Recommendations | Actual Purchases (test) |
|------|------------------------------|-----------------|-------------------------|
| 21001 | [11808, 7381, 9141, 10747] | [11772, 8471, 9576, 5351, 1690, 5603, 8015, 1465, 10872, 5934] | [11772] |

Table 2: Beauty dataset - A case study for user 21001 - Product 11772 - Ideal explainability

| Product | Candidate Explainabilities | Exp. Score |
|---------|----------------------------|------------|
| 11772 | user 21001 has purchase product 10747 which was purchase by user 9748 who purchase product 11772 | 0.8712647 |
| | user 21001 has purchase product 10747 which was also viewed by related product 23132 who also viewed product 11772 | 1.4115256 |
| | user 21001 has purchase product 10747 which was produced by brand 201 who produced product 11772 | 1.8997045 |
| | user 21001 has purchase product 11808 which was also viewed by related product 23062 who bought together product 11772 | 1.720593 |
| | user 21001 has purchase product 11808 which was also viewed by related product 23062 who also bought product 11772 | 1.720593 |
| | user 21001 has purchase product 11808 which was also viewed by related product 23062 who also viewed product 11772 | 1.720593 |

tions, the statement "User 21001 purchased product 10747, which was produced by brand 201, who also produced product 11772" achieved the highest explainability score. Consequently, the model selects the explanation with the highest score to ensure optimal transparency and user trust in the recommendations.

As previously mentioned, the objective of this paper is to introduce confidence interval-based scoring for MES, which currently operates as a single-point value. Table 3 presents the explainability for all the top-10 recommended products, including their MES values and the corresponding confidence intervals. As outlined in Table 2, the process of selecting the most meaningful explanation for a recommended product involves identifying the highest-scoring explainability. Table 3 extends this approach by displaying the explainability scores of all the top-10 recommended products alongside their MES values and their respective confidence intervals, offering a more comprehensive assessment of recommendation transparency.

The evaluation of confidence intervals follows Algorithm 1, where the model first calculates the standard deviation of all possible explainability scores. Based on the desired confidence level, it then determines the corresponding confidence interval. Table 3 presents the 95% confidence interval for the respective MES values. In some instances, the MES-CI field is blank, as there was only one single possible explanation for the recommended product in those cases. The inclusion of confidence interval values strengthens user trust by providing a clearer assessment of MES and the corresponding explainability of recommendations.

## 5.3 Limitation

While the MES-CI framework enhances explainability assessment by incorporating confidence intervals, it has certain limitations. It relies on the statistical distribution of explainability scores, assuming a well-behaved spread, which may not accurately reflect true reliability in cases of high variance or skewness. Additionally, for recommended products with only one possible explanation, MES-CI cannot provide meaningful confidence bounds, limiting interpretability in those instances. The framework is also sensitive to sample size, as a smaller number of explanations can result in wider or unstable confidence intervals, affecting robustness. Addressing these limitations could further improve the reliability and practical usability of MES-CI in XAI systems.

Table 3: Beauty dataset - A case study for user 21001 - Recommendations explainability

| Recommendation | Explainability of recommendation | MES | MES-CI |
|---|---|---|---|
| 11772 | [user 21001 has purchase product 10747 which was produced by brand 201 who produced product 11772] | 1.8997045 | [1.62, 2.178] |
| 8471 | [user 21001 has purchase product 10747 which was produced by brand 201 who produced product 8471] | 1.8298879 | [1.627, 2.032] |
| 9576 | [user 21001 has purchase product 10747 which was produced by brand 201 who produced product 9576] | 1.8205801 | [1.48, 2.16] |
| 5351 | [user 21001 has purchase product 10747 which was produced by brand 201 who produced product 5351] | 1.6967345 | [1.052, 2.341] |
| 1690 | [user 21001 has purchase product 9141 which was also bought by related product 11466 who also bought product 1690] | 1.1402445 | [] |
| 5603 | [user 21001 has mentions word 4603 which was mentions by user 5593 who purchase product 5603] | 0.33712164 | [] |
| 8015 | [user 21001 has mentions word 22373 which was mentions by user 1924 who purchase product 8015] | 0.6660103 | [] |
| 1465 | [user 21001 has purchase product 9141 which was also bought by related product 11466 who also bought product 1465] | 0.9332369 | [] |
| 10872 | [user 21001 has purchase product 9141 which was purchase by user 11576 who purchase product 10872] | 0.7701013 | [] |
| 5934 | [user 21001 has purchase product 10747 which was produced by brand 201 who produced product 5934] | 1.5355128 | [0.445, 2.625] |

# 6   Conclusion and future works

This paper builds upon the existing MES framework by introducing a confidence interval-based scoring mechanism to improve the explainability of recommendations. By incorporating confidence intervals, the proposed approach enhances user trust and provides a more nuanced evaluation of explainability beyond a single-point metric. The results demonstrate that this refined methodology offers a deeper understanding of recommendation transparency, allowing users to assess both the reliability and informational value of explanations. The approach ensures that the most meaningful explanations that maximize both rewards and information gain—are selected, thereby improving the overall effectiveness of XRS.

Future research will explore additional enhancements to the MES-CI framework to address its limitations, such as refining confidence interval calculations for sparse datasets and ensuring adaptability in dynamic user contexts. Investigating alternative statistical models for explainability quantification may further improve the robustness of confidence intervals. Additionally, extending MES-CI to multi-modal RSs and real-time personalization scenarios will help validate its applicability across diverse AI-driven environments. These efforts will contribute to the continuous advancement of transparent, interpretable, and user-centric AI systems.

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

# A Technical Appendices and Supplementary Material

This section includes technical appendices featuring supplementary results, figures, graphs, and mathematical proofs.

## A.1 Data

### A.1.1 Data Description

The experimental data was sourced from previous research [16] and is available at the provided link. The experiment employed two domain-specific datasets from the Amazon e-commerce platform, specifically the Clothing and Beauty domains. These datasets were selected due to their richness in user-product interactions and the availability of diverse product metadata, making them ideal for evaluating knowledge-aware and XRS. As illustrated in Figure 1, each dataset was modeled using a structured schema incorporating six key entity types: User, Product, Product's Feature Words, Related Products, Brand, and Category.

The relationships among these entities were both varied and semantically meaningful, encompassing eight distinct relationship types that reflected different aspects of e-commerce interactions. For instance, users interacted with products through purchases, which were further contextualized by product categories (e.g., shirts, makeup kits) and brands (e.g., Nike, L'Oréal). Additionally, each product was associated with a set of related products through behavioral patterns captured in the dataset—such as "also viewed", "also bought", or "bought together" relationships. These links provided valuable insights into co-purchasing behavior and product affinity, which are essential for collaborative filtering and path-based reasoning.

Moreover, the datasets included feature words that were extracted from user reviews and product descriptions. These words served as textual signals reflecting product characteristics and user sentiments. The connections between products and these feature words enabled the model to incorporate semantic information into the recommendation process, enhancing interpretability and enabling more personalized results.

To effectively organize and represent this complex, multi-relational data, a user-centric KG was formed. This KG helped identify and formalize the various entities and their relationships in the form of directed triples $(h, r, t)$ where $h$ represents the head entity, $r$ the relationship, and $t$ the tail entity. For example, a triple like (User A, purchased, Product B) or (Product B, produced_by, Brand C) encodes a specific piece of structured knowledge that the model can leverage. This structured representation allowed the recommender system to perform path-based reasoning, uncover latent connections, and generate recommendations that are both contextually relevant and explainable.

Overall, this phase established a rich, semantically grounded foundation that supported subsequent stages in the pipeline, including knowledge graph embedding, reinforcement learning, and the generation of transparent, personalized recommendations..

### A.1.2 Data Summary

Table 4 provides a statistical summary that offers valuable insights into the entities, relationships, and characteristics of the datasets used in the experiment, enabling a deeper understanding of the data structure. The top section of the table outlines the number of entities present within each dataset, while the bottom section focuses on the relationships between head and tail entities, presenting the mean and standard deviation values. Among the various relationships, 'mention' and 'described as' are consistently prevalent across all datasets, underscoring their significance in capturing product features. However, these relationships may contain redundant words, which can be refined using the Term Frequency - Inverse Document Frequency (TF-IDF) technique to filter out less relevant feature words. Additionally, within the Product and Related Products datasets, the 'also bought' relationship stands out as the most dominant, highlighting its strong influence in modeling user purchasing behavior.

### A.1.3 Data Snippet

Table 5 provides a detailed overview of the Amazon 'clothing' e-commerce dataset used in the experiment. The upper section summarizes key entities within the dataset, with anonymized details for users, products, and related product entities. The lower section

Table 4: Experimentation - Descriptions and statistics of Amazon e-commerce datasets

|  |  | Clothing | Beauty |
|---|---|---|---|
| Entities | Description | Number of Entities | |
| User | User in recommender system | 39,387 | 22,363 |
| Product | Product to be recommended to users | 23,033 | 12,101 |
| Feature/Word | A product feature word from reviews | 21,366 | 22,564 |
| RelatedProduct | Related bought/viewed products | 339,367 | 164,721 |
| Brand | Brand or manufacturer of the product | 1,182 | 2,077 |
| Category | Category of the product | 1,193 | 248 |
| Relations | Description | Mean and Std. Deviation per Head Entity | |
| purchase | User purchased Product | 7.1±3.6 | 8.9±8.2 |
| mention | User mentioned Feature/Word | 440.2±452.4 | 806.9±1344.1 |
| described_as | Product described as Feature/Word | 752.7±909.4 | 1,491.2±2,554 |
| belong_to | Product belong to Category | 6.7±2.1 | 4.1±0.7 |
| produced_by | Product produced by Brand | 0.2±0.4 | 0.8±0.4 |
| also_bought | Product also bought Related Product | 61.3±33 | 73.6±30.7 |
| also_viewed | Product also viewed Related Product | 6.3±6.2 | 12.8±9 |
| bought_together | Product bought together Related Product | 0.7±0.9 | 0.7±0.7 |

Table 5: Sample of the first five records from the Amazon 'clothing' e-commerce dataset

| Snapshot of the entity dataset | | | | | |
|---|---|---|---|---|---|
| Product | User | Word | Brand | Category | Related Product |
| B0000A4ZJD | A1A0CEX9QSLWQF | abalone | Boutique Cutie | Clothing, Shoes & Jewelry | 0000031852 |
| B0000A51FU | A1A0DUC0MTZJR0 | abandon | Disney | Girls | 0000031895 |
| B0000A53UX | A1A0IXLIVMW1EW | halo | Lewis N. Clark | Clothing | 0000031909 |
| B0000A522N | A1A0L70DM2OCW8 | handbag | Suunto | Active | 0000032034 |
| B0000AI44G | A1A0LP8RN93W3G | happy | Kidoozie | Active Skirts | 0000032042 |

| Snapshot of relationships, including transactional interactions from the Train and Test datasets | | | | | | | |
|---|---|---|---|---|---|---|---|
| User Purchase Product, User Mention Word, Product Described As Word | | | Product Produced Brand | Product Belongs To Category | Product Also Bought Related Product | Product Also Viewed Related Product | Product Bought Together Related Product |
| User | Product | Word | Brand | Category | Related Product | Related Product | Related Product |
| 431 | 8374 | 8028 12395 … | 8 | 0 25 423 711 39 | 157044 40113 … | 7644 7647 8868 … | 40113 148881 |
| 2925 | 8374 | 10319 15312 … | 738 | 0 129 2 164 261 | 8086 17839 18136 … | 12789 12805 34704 … | |
| 18864 | 8374 | 6465 8651 … | | 0 1 2 91 49 | 33884 73544 73560 … | 5995 5954 7777 … | |
| 7386 | 8374 | 3016 18525 … | 433 | 0 28 5 14 15 | 37152 37072 37102 … | 4184 380 390 …. | |
| 16467 | 8374 | 4142 1127 … | 30 | 0 357 48 25 62 5 | 63213 217158 … | 10921 4840 4803 … | 93382 93395 |

highlights the three most significant relationships in the dataset, helping to contextualize interactions between entities. For instance, User #431, represented in the user entity, purchases Product #8374 from the product entity, forming the relationship 'user purchasing product.' Additionally, User #431 references specific feature words such as #8028 and #12395, which are recorded in the word entity while purchasing Product #8374. Similarly, Product #8374 is linked to multiple feature words, including #8028, #12395, and others, establishing the 'product described by words' relationship. The subsequent columns illustrate associations between products and various entities, including brand, category, and related products. Each row corresponds to a sequential product identifier. For example, Product #1 in the product entity is manufactured by Brand #8 in the brand entity and classified under Categories #0, #25, #423, #711, and #39 in the category entity.

Additionally, #Product_1 is frequently purchased under the "Also Bought" relationship alongside related products such as #157044, #40113, and many others. Furthermore, #Product_1 is often viewed under the "Also Viewed" category alongside related products like #7644, #7647, #8868, and others. Lastly, #Product_1 is commonly purchased together with related products such as #40113 and #148881.

This structured representation highlights the intricate network of connections within the Amazon 'clothing' e-commerce dataset, illustrating the dynamic interplay between entities

Table 6: Experimental Environment Settings

| Environment | Type | Value | Version |
|---|---|---|---|
| H/W & OS | OS | Microsoft Windows 11 Home | 10.0.26100 Build 26100 |
| | Processor | i9-14900HX, | 24 cores |
| | RAM | 32 GB | |
| | Storage | SSD | 1 TB |
| | GPU | Nvidia GeForce RTX4060 | |
| S/W & Programming | Programming Platform | VSCode | 1.100.1 |
| | Programming Language | Python | 3.12 |
| | Deep Learning Framework | Pytorch | 1.13.1 |

and their relationships. Essentially, entities function as dimensions, while relationships serve as factual links that connect these dimensions.

### A.2 Experiment

#### A.2.1 Experimental Environment

For a comprehensive overview of the experiment's specifics, detailed information can be found in Table 6, where the experimental settings are documented.

#### A.2.2 Experimental Setup

The research utilized a comprehensive transaction dataset that was systematically divided into training and test subsets. Each subset encompassed critical information such as users' product purchase histories, the occurrence of feature-related word mentions, and detailed product descriptions. This structured data enabled the development of a semantically enriched RS. Table 7 presents a summary of statistical information, including the mean and standard deviation for each subset across two distinct domains: Clothing, and Beauty. These statistics offer insight into user behavior and interaction patterns, highlighting variations in the number of products purchased and the frequency of feature word mentions.

For example, in the "Clothing" domain, users purchased an average of 7.08 products with a standard deviation of 3.59, reflecting a moderate variation in user activity. When broken down by dataset split, users in the training set purchased an average of 5.45 products (SD = 2.49), whereas those in the test set purchased 1.62 products on average (SD = 1.13). This disparity underscores the segmentation strategy, which simulates real-world scenarios where models are trained on historical data and evaluated on unseen user behavior. Similar trends were observed across the other domains, providing a diverse and robust experimental foundation.

The experimental framework was implemented using Python as the primary programming language, with PyTorch employed as the deep learning library due to its flexibility and efficiency in building complex neural models. The proposed RS integrated KG and RL technologies to generate semantically informed and context-aware recommendations. The KG served to represent structured domain knowledge, capturing intricate relationships between users, products, and attributes. RL was used to dynamically model user preferences over time, enabling the system to learn optimal recommendation strategies through interaction and feedback.

This hybrid approach aligns with the broader objectives of XAI, as it facilitates transparency in recommendation reasoning by leveraging semantic relationships and adaptive decision-making. To evaluate the effectiveness of the RS, standard top-N recommendation metrics were employed, including NDCG, Recall, HR, and Precision. These metrics provide a comprehensive assessment of the system's ability to rank relevant items accurately, capture user interests, and deliver consistent performance across varied domains.

### A.3 Experiment Methodology

#### A.3.1 Data Preprocessing

The data preprocessing phase plays a critical role in preparing the dataset for semantic recommendation. This stage involves systematically extracting key components such as

Table 7: Statistics of train & test datasets for the four Amazon e-commerce datasets

| Relations | Description | Clothing | Beauty |
|---|---|---|---|
| | | Number of Records / Transactions | |
| Total Records | # transactions | 278677 (100%) | 198502 (100%) |
| | Train dataset | 214696 (77.04%) | 149844 (75.49%) |
| | Test dataset | 63981 (22.96%) | 48658 (24.51%) |
| | | (Mean ± Std. Deviation) of Relations per Head Entity | |
| User purchase Product | Overall Statistics | 7.08± 3.59 | 8.88±8.16 |
| | Train Statistics | 5.45± 2.49 | 6.7±5.7 |
| | Test Statistics | 1.62± 1.13 | 2.18±2.48 |
| User mention Word | Overall Statistics | 440.2± 452.38 | 806.89±1344.08 |
| | Train Statistics | 338.1± 334.91 | 605.01±957.5 |
| | Test Statistics | 102.1± 134.21 | 201.88±401.51 |
| Product described_as Word | Overall Statistics | 752.75± 909.42 | 1491.16±2553.93 |
| | Train Statistics | 578.23± 708.51 | 1118.18±1905.02 |
| | Test Statistics | 201.19± 255.96 | 419.34±731.15 |

relevant entities (e.g., products, attributes), relationships (e.g., user-product interactions, product-feature associations), and detailed user behavior data. To ensure that the input data is both meaningful and manageable, a robust filtering mechanism is employed.

Specifically, the filtering criterion focuses on retaining only those feature words that contribute valuable semantic information. Feature words that appear more than 5,000 times are excluded, as they are likely to be overly common or generic (e.g., "good," "product," "buy") and thus contribute little to the specificity required for effective recommendations. Additionally, to further refine the dataset, only words with a Term Frequency-Inverse Document Frequency (TF-IDF) score greater than 0.2 are retained. This threshold ensures that the selected words are not only infrequent but also contextually important across the document corpus, highlighting their relevance to specific products or user preferences.

Through this process, the final dataset is enriched with significant and pertinent feature words, which enhances the quality of the KG and ultimately improves the performance and explainability of the RS.

### A.3.2    Knowledge Graph Generation

In this phase, a critical step involves constructing a Label Property Graph (LPG) from the training dataset. The LPG serves as a foundational data structure that models the semantic and relational aspects of the domain. It is built by extracting and organizing key entities—such as users, products, and feature words—along with their bi-directional relationships and interactions. These relationships include user-product interactions (e.g., purchases or ratings), product-feature associations, and semantic links between products and descriptive terms. By incorporating directionality, the graph captures not only the existence of relationships but also the flow of influence or interaction between entities, enhancing the expressiveness of the model.

Each node in the LPG is assigned a label (e.g., "User", "Product", "Feature") and associated properties (such as user ID, product category, TF-IDF score, or purchase frequency), enabling a structured and attribute-rich representation of the dataset. This labeling schema helps in differentiating node types during the learning process and allows the model to leverage domain semantics effectively. The construction of the LPG thus provides a comprehensive framework for encoding and navigating the underlying knowledge within the data.

Although the LPG is primarily generated from the training dataset, both the training and test datasets contribute to the definition of labels and properties, ensuring consistency across the entire evaluation pipeline. This alignment is essential, as it enables the recommender system to generalize learned representations from the training phase and apply them effectively to unseen data during testing. Furthermore, the LPG plays a vital role in supporting explainability by maintaining interpretable relationships and features that can be traced back to individual recommendations. As a result, the LPG not only enhances model performance but also facilitates transparency and trust in the recommendation process, aligning with the goals of semantic modeling and XAI.

### A.3.3 Knowledge Graph Embedding

The third phase of the experiment focuses on learning low-dimensional vector representations of entities and relationships within the KG using the TransE (Translating Embeddings) algorithm. TransE is a widely used KGE technique that models relationships by interpreting them as translations in the vector space. In this approach, for a given triplet $(h, r, t)$—where $h$ is the head entity, $r$ is the relation, and $t$ is the tail entity—TransE seeks to learn embeddings such that the vector representation of the head entity plus the relation vector approximates the tail entity vector, i.e., $\mathbf{h} + \mathbf{r} \approx \mathbf{t}$.

To manage computational efficiency and scalability, embedding vectors are generated in mini-batches. For the Beauty, and Cell Phones domains, a batch size of 32 is selected to balance memory consumption and convergence speed. This batching strategy ensures that the learning process remains efficient across datasets of varying scales.

The embedding process is carried out over 30 epochs, during which the model iteratively updates the representations of entities and relationships based on observed interactions and structural dependencies in the graph. During each epoch, the model adjusts its internal parameters by minimizing a smooth margin-based ranking loss function, which encourages correct triplets to have lower distances between embeddings than negative triplets. This loss function plays a key role in shaping meaningful geometric relationships in the embedding space, capturing semantic similarities and structural patterns.

By the end of this phase, the embedding model yields a set of well-trained vectors that encode the semantic proximity and relational dynamics among entities in the KG. These learned embeddings are essential for downstream tasks, such as making recommendations and generating explanations, as they provide a compact, information-rich representation of the complex interconnections within the dataset. This phase lays the groundwork for integrating semantic understanding into the recommender system and supports the broader goal of enhancing explainability in AI-driven recommendations.

### A.3.4 Reinforcement Learning Model

In this phase, the focus shifts to analyzing users' historical purchase behaviors and leveraging this information to train a RL agent capable of making personalized product recommendations. The model is designed to explore the KG by simulating a sequence of decisions that reflect a user's potential navigation paths through the graph. Each path represents a trajectory from a user node to a product node, capturing the semantic associations and behavioral patterns that inform recommendation outcomes.

To maintain computational tractability and ensure semantic relevance, the exploration of the KG is constrained to a maximum of three hops. Each hop allows the agent to choose among several candidate actions—such as transitioning to a related product, feature, or category node—based on the current node and its neighboring relationships. These actions determine the agent's trajectory and influence the subsequent choices available in future hops. Importantly, even if the agent ends its traversal at a product node, there is no guarantee that the node corresponds to a product previously seen in the training or test set for that user, introducing a realistic challenge of exploring novel but relevant items.

The RL agent operates in a step-wise learning framework, where it continuously updates its policy and parameter weights at each decision point rather than deferring learning until the end of the episode. This temporal difference learning approach allows the agent to react dynamically to immediate feedback and refine its decision-making strategy over time. The agent receives positive rewards for successfully navigating paths that lead to target-labeled products—i.e., products aligned with the user's known interests or preferences. These rewards signal successful behavior, guiding the agent to prioritize similar paths in the future.

The RL algorithm is carefully designed to incentivize the discovery and exploitation of user behavior patterns. By observing and internalizing how users interact with different product attributes and categories, the agent builds a behavioral model that encapsulates user preferences. This model becomes highly valuable during the recommendation phase, as it enables the agent to proactively identify products that are not only semantically related to the user's historical interests but are also behaviorally consistent with their past choices. Over time, the RL agent learns to balance exploration (discovering new products) and exploitation (recommending known preferences), ultimately enhancing the accuracy, relevance, and explainability of the RS.

### A.3.5 Experimental Parameter Settings and Evaluation Strategy

The experimental setup adopted a well-defined set of *default parameter settings* to establish a consistent and reproducible baseline for evaluating the model's performance across multiple datasets. These parameters were carefully selected based on empirical observations and prior research to balance computational efficiency with model accuracy and explainability.

To generate *latent representations*, the models employed knowledge graph embeddings trained using a *1-hop scoring function*, enabling the system to capture meaningful local semantics between entities and relationships. Each embedding vector had a dimensionality of 100, providing sufficient capacity to represent nuanced interactions within the knowledge graph. The embedding training process used a learning rate of 0.5 and was capped at a maximum of 30 epochs to prevent overfitting while ensuring convergence.

The *Actor-Critic reinforcement learning (RL) model* was optimized for interpretability and path efficiency, particularly by constraining the *maximum path length to three hops*. This design choice aimed to generate concise and intuitive explanation paths that end users can easily comprehend. To manage computational complexity and maintain recommendation quality, the *action space* was pruned to a maximum of 250 actions per state. An *action dropout rate of 0.5* was applied during training to encourage the agent to explore diverse paths, thereby enhancing both learning generalization and the diversity of explanations.

Key RL parameters included a *discount factor* $\gamma = 0.99$, reflecting a high degree of importance placed on future rewards, which is crucial for long-term planning in sequential decision-making tasks. The policy and value networks were defined with the following weight matrix configurations:

- $\mathbf{W}_1 \in \mathbb{R}^{400 \times 512}$: Input to first hidden layer
- $\mathbf{W}_2 \in \mathbb{R}^{512 \times 256}$: First hidden to second hidden layer
- $\mathbf{W}_P \in \mathbb{R}^{256 \times 250}$: Policy head (action probability)
- $\mathbf{W}_V \in \mathbb{R}^{256 \times 1}$: Value head (expected return)

Training procedures were dataset-specific. For the *CDs & Vinyl* dataset, the model was trained for 5 epochs using the *Adam optimizer* with a learning rate of 0.001 and a batch size of 64. For other datasets (e.g., Clothing, Beauty), the model was trained for 100 epochs with a lower learning rate of 0.0001 and a batch size of 32 to accommodate differences in dataset scale and convergence behavior. The entropy loss weight was set to 0.001, promoting exploration by penalizing overly confident predictions.

To ensure *continuity and reproducibility*, both the embedding and RL models were configured to resume training from the most recent checkpoint, enabling seamless recovery and consistent results.

During the *evaluation and product recommendation phase*, the Actor-Critic model was evaluated across all users in each dataset using default parameter settings. For three-hop sampling, the default parameters were:

$$K_1 = 10, \quad K_2 = 10, \quad K_3 = 12$$

where $K_i$ denotes the number of candidate actions at hop $i$.

The evaluation phase also incorporated the *Maximum Explainability Score with Confidence Interval (MES-CI)* module to further refine the quality of recommendations. MES-CI facilitated the selection of semantic paths that optimized not only the relevance of recommendations but also their interpretability and user trust, thereby aligning the system with the broader goals of explainable and user-centric AI.

### A.4 Recommendation

The final phase of the experiment is dedicated to evaluating the explainability of the proposed RS. This phase is critical as it not only measures the accuracy and effectiveness of the model's predictions but also ensures that the recommendations are interpretable and aligned with the principles of XAI. The evaluation is conducted in two main steps: the Prediction Step and the Recommendation Step.

- **Prediction step**: In this step, the model estimates a set of key parameters for each candidate product that could potentially be recommended to a user. These parameters are derived from the semantic paths explored in the KG and provide insights into both the system's reasoning and the user's behavior. The four predicted parameters are:

Table 8: Execution time - with GPU enabled device

| Process Step | Command | Execution Time |
|---|---|---|
| Proprocess the data first: | python preprocess.py –dataset <dataset_name> | 2-7 Hours |
| Train knowledge graph embeddings | python train_transe_model.py –dataset <dataset_name> | 2-7 days |
| Train RL agent | python train_RL_agent.py –dataset <dataset_name> | 2-7 days |
| Evaluation - MESCI - Generic call | python test_RL_agent.py –dataset <dataset_name> – run_path True –run_eval True –users <user_id> –debug True | 2-7 mins |
| Performance Evaluation | python test_RL_agent.py –dataset <dataset_name> – run_path True –run_eval True | 2-5 hours |

- **Likelihood of Traversal (P)**: This refers to the probability that a user will traverse a specific semantic path in the KG. It reflects the system's confidence that the user would follow a given sequence of relationships and entities based on historical behavior.
- **Product Affinity Score (S)**: This score represents the predicted degree of interest or preference a user has for a particular product. It is derived from the user's previous interactions, preferences for features, and relationships encoded in the graph.
- **Cumulative Reward (Rw)**: This metric captures the total reward the RL agent expects to gain by following a specific path that ends in a candidate product. It quantifies how well a given path aligns with the user's known preferences, as learned during training.
- **Path Entropy (H)**: Entropy reflects the level of uncertainty associated with a given path. A high entropy value suggests a lack of clarity or confidence in the path's outcome, while a low entropy indicates a more deterministic and explainable path. This helps in filtering out ambiguous or less reliable recommendation paths.

- **Recommendation step**: Once the prediction parameters are obtained, the model proceeds to the Recommendation Step, where it utilizes these parameters to determine the optimal explainability path for each user-product pair. This is accomplished using the Maximum Explainability Score with Confidence Interval (MES-CI) algorithm. The MES-CI algorithm balances three competing objectives—accuracy, confidence, and informativeness—to select paths that are not only effective in predicting user preferences but also transparent and easy to interpret.

After identifying the optimal semantic path, the system recommends the associated product(s) to the user. Importantly, the selected path also serves as the explanation for why the product is being recommended. For instance, the system might explain that a product is recommended because it shares features with items the user previously purchased, comes from a favored brand, or is frequently bought together with past purchases.

By incorporating explainability into both the prediction and recommendation processes, the RS ensures that its outputs are not only personalized and accurate, but also transparent and trustworthy. This dual emphasis enhances user satisfaction and trust, and aligns the system with broader XAI goals, promoting adoption in real-world applications where understanding the rationale behind AI decisions is critical.

### A.5  Execution Steps:

***Max Explainability Score with Confidence Interval (MES-CI): A Quantitative Metric for Interpretability in Knowledge Graph-Based Recommender Systems***

*Datasets* Datasets used in this paper is available at the provided link

The execution time is listed in the Table 8.

*Requirements*

- Python >= 3.12

- PyTorch> = 2.5

***How to run the code***

1. Proprocess the data first:

```bash
python preprocess.py –dataset <dataset_name>
```

```
"""
```

"<dataset_name>" should be one of "beauty", "cloth" (refer to utils.py).

2. Train knowledge graph embeddings (TransE in this case):

```bash
python train_transe_model.py –dataset <dataset_name>
```

3. Train RL agent:

```bash
python train_RL_agent.py –dataset <dataset_name>
```

4a. Evaluation - MESCI - Example call

```bash
python test_RL_agent.py –dataset beauty –run_path True –run_eval True –users 21001 –debug True
```

If "run_path" is True, the program will generate paths for recommendation according to the trained policy.

If "run_eval" is True, the program will evaluate the recommendation performance based on the resulting paths.

4b. Evaluation - MESCI - Generic call

```bash
python test_RL_agent.py –dataset <dataset_name> –run_path True –run_eval True –users <user_id> –debug True
```

If "run_path" is True, the program will generate paths for recommendation according to the trained policy.

If "run_eval" is True, the program will evaluate the recommendation performance based on the resulting paths.

4c. Performance Evaluation

```bash
python test_RL_agent.py –dataset <dataset_name> –run_path True –run_eval True
```

If "run_path" is True, the program will generate paths for recommendation according to the trained policy.

If "run_eval" is True, the program will evaluate the recommendation performance based on the resulting paths.

## A.6 Logs-Amazon Beauty

### A.6.1 Training - Embedding Model

[INFO] Namespace(dataset='beauty', name='train_transe_model', seed=123, gpu='1', epochs=30, batch_size=16, lr=0.5, weight_decay=0, l2_lambda=0, max_grad_norm=5.0, embed_size=100, num_neg_samples=5, steps_per_checkpoint=100000, checkpoint_folder='checkpoint', log_folder='log', log_file_name='train_log.txt', is_resume_from_checkpoint=0, logging_mode='a', device=device(type='cuda', index=0), dir='./tmp/Amazon_Beauty/train_transe_model', checkpoint_dir='./tmp/Amazon_Beauty/train_transe_model/checkpoint', log_dir='./tmp/Amazon_Beauty/train_transe_model/log')

[INFO] Parameters:['purchase', 'mentions', 'describe_as', 'produced_by', 'belongs_to', 'also_bought', 'also_viewed', 'bought_together', 'user.weight', 'product.weight', 'word.weight', 'related_product.weight', 'brand.weight', 'category.weight', 'purchase_bias.weight', 'mentions_bias.weight', 'describe_as_bias.weight', 'produced_by_bias.weight', 'belongs_to_bias.weight', 'also_bought_bias.weight', 'also_viewed_bias.weight', 'bought_together_bias.weight']

[INFO] Epoch: 01 | Words: 4128430/405897991 | Lr: 0.49491 | Smooth loss: 14.59327
[INFO] Epoch: 01 | Words: 8258715/405897991 | Lr: 0.48983 | Smooth loss: 12.55730
[INFO] Epoch: 01 | Words: 12396929/405897991 | Lr: 0.48473 | Smooth loss: 12.04881

[INFO] Epoch: 02 | Words: 16532862/405897991 | Lr: 0.47963 | Smooth loss: 11.54240
[INFO] Epoch: 02 | Words: 20672014/405897991 | Lr: 0.47454 | Smooth loss: 11.39307
[INFO] Epoch: 02 | Words: 24796721/405897991 | Lr: 0.46945 | Smooth loss: 11.35870
[INFO] Epoch: 03 | Words: 28926876/405897991 | Lr: 0.46437 | Smooth loss: 11.08418
[INFO] Epoch: 03 | Words: 33060941/405897991 | Lr: 0.45927 | Smooth loss: 10.84952
[INFO] Epoch: 03 | Words: 37189105/405897991 | Lr: 0.45419 | Smooth loss: 10.87816
[INFO] Epoch: 04 | Words: 41320252/405897991 | Lr: 0.44910 | Smooth loss: 10.80380
[INFO] Epoch: 04 | Words: 45447898/405897991 | Lr: 0.44402 | Smooth loss: 10.48704
[INFO] Epoch: 04 | Words: 49586669/405897991 | Lr: 0.43892 | Smooth loss: 10.52798
[INFO] Epoch: 04 | Words: 53717858/405897991 | Lr: 0.43383 | Smooth loss: 10.51342
[INFO] Epoch: 05 | Words: 57854228/405897991 | Lr: 0.42873 | Smooth loss: 10.20127
[INFO] Epoch: 05 | Words: 61980330/405897991 | Lr: 0.42365 | Smooth loss: 10.18586
[INFO] Epoch: 05 | Words: 66114677/405897991 | Lr: 0.41856 | Smooth loss: 10.21038
[INFO] Epoch: 06 | Words: 70248616/405897991 | Lr: 0.41347 | Smooth loss: 9.99730
[INFO] Epoch: 06 | Words: 74383499/405897991 | Lr: 0.40837 | Smooth loss: 9.92230
[INFO] Epoch: 06 | Words: 78516640/405897991 | Lr: 0.40328 | Smooth loss: 9.92832
[INFO] Epoch: 07 | Words: 82652426/405897991 | Lr: 0.39819 | Smooth loss: 9.79496
[INFO] Epoch: 07 | Words: 86786576/405897991 | Lr: 0.39309 | Smooth loss: 9.63500
[INFO] Epoch: 07 | Words: 90915476/405897991 | Lr: 0.38801 | Smooth loss: 9.66572
[INFO] Epoch: 08 | Words: 95050419/405897991 | Lr: 0.38291 | Smooth loss: 9.63382
[INFO] Epoch: 08 | Words: 99180036/405897991 | Lr: 0.37783 | Smooth loss: 9.37495
[INFO] Epoch: 08 | Words: 103312880/405897991 | Lr: 0.37274 | Smooth loss: 9.38763
[INFO] Epoch: 08 | Words: 107450397/405897991 | Lr: 0.36764 | Smooth loss: 9.39443
[INFO] Epoch: 09 | Words: 111579451/405897991 | Lr: 0.36255 | Smooth loss: 9.17835
[INFO] Epoch: 09 | Words: 115711107/405897991 | Lr: 0.35746 | Smooth loss: 9.18696
[INFO] Epoch: 09 | Words: 119848199/405897991 | Lr: 0.35237 | Smooth loss: 9.17020
[INFO] Epoch: 10 | Words: 123980140/405897991 | Lr: 0.34728 | Smooth loss: 9.02826
[INFO] Epoch: 10 | Words: 128115148/405897991 | Lr: 0.34218 | Smooth loss: 8.93950
[INFO] Epoch: 10 | Words: 132244427/405897991 | Lr: 0.33710 | Smooth loss: 8.91189
[INFO] Epoch: 11 | Words: 136375962/405897991 | Lr: 0.33201 | Smooth loss: 8.86128
[INFO] Epoch: 11 | Words: 140511594/405897991 | Lr: 0.32691 | Smooth loss: 8.70399
[INFO] Epoch: 11 | Words: 144644269/405897991 | Lr: 0.32182 | Smooth loss: 8.71882
[INFO] Epoch: 11 | Words: 148774311/405897991 | Lr: 0.31673 | Smooth loss: 8.70265
[INFO] Epoch: 12 | Words: 152907720/405897991 | Lr: 0.31164 | Smooth loss: 8.47386
[INFO] Epoch: 12 | Words: 157039204/405897991 | Lr: 0.30655 | Smooth loss: 8.52233
[INFO] Epoch: 12 | Words: 161168616/405897991 | Lr: 0.30147 | Smooth loss: 8.50717
[INFO] Epoch: 13 | Words: 165296435/405897991 | Lr: 0.29638 | Smooth loss: 8.34437
[INFO] Epoch: 13 | Words: 169429053/405897991 | Lr: 0.29129 | Smooth loss: 8.30853
[INFO] Epoch: 13 | Words: 173561225/405897991 | Lr: 0.28620 | Smooth loss: 8.29629
[INFO] Epoch: 14 | Words: 177694465/405897991 | Lr: 0.28111 | Smooth loss: 8.19751
[INFO] Epoch: 14 | Words: 181820934/405897991 | Lr: 0.27603 | Smooth loss: 8.10413
[INFO] Epoch: 14 | Words: 185954101/405897991 | Lr: 0.27093 | Smooth loss: 8.11419
[INFO] Epoch: 15 | Words: 190086525/405897991 | Lr: 0.26584 | Smooth loss: 8.05563
[INFO] Epoch: 15 | Words: 194219304/405897991 | Lr: 0.26075 | Smooth loss: 7.90554
[INFO] Epoch: 15 | Words: 198349234/405897991 | Lr: 0.25567 | Smooth loss: 7.91409
[INFO] Epoch: 15 | Words: 202485461/405897991 | Lr: 0.25057 | Smooth loss: 7.92219
[INFO] Epoch: 16 | Words: 206619102/405897991 | Lr: 0.24548 | Smooth loss: 7.73590
[INFO] Epoch: 16 | Words: 210751866/405897991 | Lr: 0.24039 | Smooth loss: 7.74630
[INFO] Epoch: 16 | Words: 214884114/405897991 | Lr: 0.23530 | Smooth loss: 7.74051
[INFO] Epoch: 17 | Words: 219023607/405897991 | Lr: 0.23020 | Smooth loss: 7.62290
[INFO] Epoch: 17 | Words: 223159476/405897991 | Lr: 0.22510 | Smooth loss: 7.56469
[INFO] Epoch: 17 | Words: 227291204/405897991 | Lr: 0.22001 | Smooth loss: 7.56370
[INFO] Epoch: 18 | Words: 231420203/405897991 | Lr: 0.21493 | Smooth loss: 7.48413
[INFO] Epoch: 18 | Words: 235552208/405897991 | Lr: 0.20984 | Smooth loss: 7.40282
[INFO] Epoch: 18 | Words: 239684246/405897991 | Lr: 0.20475 | Smooth loss: 7.39227
[INFO] Epoch: 19 | Words: 243814072/405897991 | Lr: 0.19966 | Smooth loss: 7.36665
[INFO] Epoch: 19 | Words: 247936836/405897991 | Lr: 0.19458 | Smooth loss: 7.23048
[INFO] Epoch: 19 | Words: 252069613/405897991 | Lr: 0.18949 | Smooth loss: 7.23733
[INFO] Epoch: 19 | Words: 256208565/405897991 | Lr: 0.18439 | Smooth loss: 7.22914

[INFO] Epoch: 20 | Words: 260339175/405897991 | Lr: 0.17930 | Smooth loss: 7.08637
[INFO] Epoch: 20 | Words: 264475017/405897991 | Lr: 0.17421 | Smooth loss: 7.06966
[INFO] Epoch: 20 | Words: 268610900/405897991 | Lr: 0.16912 | Smooth loss: 7.08132
[INFO] Epoch: 21 | Words: 272747109/405897991 | Lr: 0.16402 | Smooth loss: 6.99974
[INFO] Epoch: 21 | Words: 276879836/405897991 | Lr: 0.15893 | Smooth loss: 6.94533
[INFO] Epoch: 21 | Words: 281015129/405897991 | Lr: 0.15384 | Smooth loss: 6.93809
[INFO] Epoch: 22 | Words: 285146733/405897991 | Lr: 0.14875 | Smooth loss: 6.89355
[INFO] Epoch: 22 | Words: 289274245/405897991 | Lr: 0.14366 | Smooth loss: 6.78789
[INFO] Epoch: 22 | Words: 293414551/405897991 | Lr: 0.13856 | Smooth loss: 6.79410
[INFO] Epoch: 22 | Words: 297544067/405897991 | Lr: 0.13347 | Smooth loss: 6.79358
[INFO] Epoch: 23 | Words: 301672950/405897991 | Lr: 0.12839 | Smooth loss: 6.65336
[INFO] Epoch: 23 | Words: 305807581/405897991 | Lr: 0.12330 | Smooth loss: 6.66337
[INFO] Epoch: 23 | Words: 309938681/405897991 | Lr: 0.11821 | Smooth loss: 6.66777
[INFO] Epoch: 24 | Words: 314075645/405897991 | Lr: 0.11311 | Smooth loss: 6.55897
[INFO] Epoch: 24 | Words: 318205910/405897991 | Lr: 0.10802 | Smooth loss: 6.52844
[INFO] Epoch: 24 | Words: 322338934/405897991 | Lr: 0.10293 | Smooth loss: 6.51966
[INFO] Epoch: 25 | Words: 326472050/405897991 | Lr: 0.09784 | Smooth loss: 6.48560
[INFO] Epoch: 25 | Words: 330602479/405897991 | Lr: 0.09275 | Smooth loss: 6.40667
[INFO] Epoch: 25 | Words: 334729687/405897991 | Lr: 0.08767 | Smooth loss: 6.41216
[INFO] Epoch: 26 | Words: 338864822/405897991 | Lr: 0.08257 | Smooth loss: 6.38024
[INFO] Epoch: 26 | Words: 342994222/405897991 | Lr: 0.07749 | Smooth loss: 6.28154
[INFO] Epoch: 26 | Words: 347126482/405897991 | Lr: 0.07240 | Smooth loss: 6.27228
[INFO] Epoch: 26 | Words: 351264737/405897991 | Lr: 0.06730 | Smooth loss: 6.28199
[INFO] Epoch: 27 | Words: 355395433/405897991 | Lr: 0.06221 | Smooth loss: 6.18766
[INFO] Epoch: 27 | Words: 359526766/405897991 | Lr: 0.05712 | Smooth loss: 6.17767
[INFO] Epoch: 27 | Words: 363661199/405897991 | Lr: 0.05203 | Smooth loss: 6.16750
[INFO] Epoch: 28 | Words: 367791726/405897991 | Lr: 0.04694 | Smooth loss: 6.11167
[INFO] Epoch: 28 | Words: 371915816/405897991 | Lr: 0.04186 | Smooth loss: 6.08268
[INFO] Epoch: 28 | Words: 376056291/405897991 | Lr: 0.03676 | Smooth loss: 6.08980
[INFO] Epoch: 29 | Words: 380192279/405897991 | Lr: 0.03167 | Smooth loss: 6.04666
[INFO] Epoch: 29 | Words: 384321214/405897991 | Lr: 0.02658 | Smooth loss: 5.98301
[INFO] Epoch: 29 | Words: 388451584/405897991 | Lr: 0.02149 | Smooth loss: 5.97555
[INFO] Epoch: 30 | Words: 392585067/405897991 | Lr: 0.01640 | Smooth loss: 5.95542
[INFO] Epoch: 30 | Words: 396714676/405897991 | Lr: 0.01131 | Smooth loss: 5.90171
[INFO] Epoch: 30 | Words: 400850887/405897991 | Lr: 0.00622 | Smooth loss: 5.88986
[INFO] Epoch: 30 | Words: 404982015/405897991 | Lr: 0.00113 | Smooth loss: 5.87715

### A.6.2   Training - RL Agentl

[INFO]    Namespace(dataset='beauty',    name='train_RL_agent',    seed=123,    gpu='1',
epochs=100,   batch_size=32,   lr=0.0001,   max_acts=250,   max_path_len=3,   gamma=0.99,
ent_weight=0.001,    act_dropout=0,    state_history=1,    hidden=[512,    256],    debug=0,
steps_per_checkpoint=50000,    checkpoint_folder='checkpoint',    log_folder='log',
log_file_name='train_log.txt',    is_resume_from_checkpoint=1,    logging_mode='a',    de-
vice=device(type='cuda',    index=0),    dir='./tmp/Amazon_Beauty/train_RL_agent',
checkpoint_dir='./tmp/Amazon_Beauty/train_RL_agent/checkpoint',
log_dir='./tmp/Amazon_Beauty/train_RL_agent/log')

[INFO] Parameters:['l1.weight', 'l1.bias', 'l2.weight', 'l2.bias', 'actor.weight', 'actor.bias',
'critic.weight', 'critic.bias']

[INFO] Save model to ./tmp/Amazon_Beauty/train_RL_agent/checkpoint/policy_model_epoch_1.ckpt

[INFO] epoch/step=2/50000 | loss=14.78206 | ploss=14.78685 | vloss=9267978.91805 | entropy=-4.79005 | reward=99.75579

[INFO] epoch/step=2/100000 | loss=-4.97828 | ploss=-4.97337 | vloss=6457813.60289 | entropy=-4.91355 | reward=70.13095

[INFO] Save model to ./tmp/Amazon_Beauty/train_RL_agent/checkpoint/policy_model_epoch_2.ckpt

[INFO] epoch/step=3/150000 | loss=-5.12558 | ploss=-5.12053 | vloss=5468618.27898 | entropy=-5.05146 | reward=59.12274

[INFO] epoch/step=3/200000 | loss=-5.14505 | ploss=-5.13992 | vloss=4560310.82483 | entropy=-5.12293 | reward=49.15593

[INFO] Save model to ./tmp/Amazon_Beauty/train_RL_agent/checkpoint/policy_model_epoch_3.ckpt

[INFO] epoch/step=4/250000 | loss=-5.16589 | ploss=-5.16076 | vloss=4646486.56127 | entropy=-5.12410 | reward=50.02885

[INFO] epoch/step=4/300000 | loss=-5.17262 | ploss=-5.16749 | vloss=4559145.30984 | entropy=-5.12619 | reward=49.15806

[INFO] epoch/step=4/350000 | loss=-5.16606 | ploss=-5.16093 | vloss=4149330.58577 | entropy=-5.12956 | reward=45.15075

[INFO] Save model to ./tmp/Amazon_Beauty/train_RL_agent/checkpoint/policy_model_epoch_4.ckpt

[INFO] epoch/step=5/400000 | loss=-5.17229 | ploss=-5.16716 | vloss=4365283.22084 | entropy=-5.13016 | reward=47.26219

[INFO] epoch/step=5/450000 | loss=-5.18155 | ploss=-5.17642 | vloss=4238572.06838 | entropy=-5.12838 | reward=46.02650

[INFO] Save model to ./tmp/Amazon_Beauty/train_RL_agent/checkpoint/policy_model_epoch_5.ckpt

[INFO] epoch/step=6/500000 | loss=-5.16002 | ploss=-5.15489 | vloss=4614346.90210 | entropy=-5.12585 | reward=49.63102

[INFO] epoch/step=6/550000 | loss=-5.17236 | ploss=-5.16723 | vloss=4599104.95268 | entropy=-5.12649 | reward=49.63485

[INFO] epoch/step=6/600000 | loss=-5.17471 | ploss=-5.16958 | vloss=4400978.84477 | entropy=-5.13000 | reward=47.57814

[INFO] Save model to ./tmp/Amazon_Beauty/train_RL_agent/checkpoint/policy_model_epoch_6.ckpt

[INFO] epoch/step=7/650000 | loss=-5.17958 | ploss=-5.17446 | vloss=3835437.65737 | entropy=-5.12670 | reward=42.07249

[INFO] epoch/step=7/700000 | loss=-5.17256 | ploss=-5.16743 | vloss=4801322.26983 | entropy=-5.13266 | reward=51.51444

[INFO] Save model to ./tmp/Amazon_Beauty/train_RL_agent/checkpoint/policy_model_epoch_7.ckpt

[INFO] epoch/step=8/750000 | loss=-5.18149 | ploss=-5.17636 | vloss=4693538.64365 | entropy=-5.12689 | reward=50.37454

[INFO] epoch/step=8/800000 | loss=-5.17321 | ploss=-5.16808 | vloss=4749648.80412 | entropy=-5.12757 | reward=51.06153

[INFO] epoch/step=8/850000 | loss=-5.18296 | ploss=-5.17783 | vloss=4029898.27107 | entropy=-5.12908 | reward=43.90774

[INFO] Save model to ./tmp/Amazon_Beauty/train_RL_agent/checkpoint/policy_model_epoch_8.ckpt

[INFO] epoch/step=9/900000 | loss=-5.19064 | ploss=-5.18551 | vloss=4342719.71911 | entropy=-5.12825 | reward=47.06476

[INFO] epoch/step=9/950000 | loss=-5.17315 | ploss=-5.16802 | vloss=4515754.65580 | entropy=-5.12752 | reward=48.76282

[INFO] Save model to ./tmp/Amazon_Beauty/train_RL_agent/checkpoint/policy_model_epoch_9.ckpt

[INFO] epoch/step=10/1000000 | loss=-5.17604 | ploss=-5.17090 | vloss=4408478.31738 | entropy=-5.13275 | reward=47.56510

[INFO] epoch/step=10/1050000 | loss=-5.17699 | ploss=-5.17187 | vloss=4391675.82434 | entropy=-5.12769 | reward=47.63739

[INFO] Save model to ./tmp/Amazon_Beauty/train_RL_agent/checkpoint/policy_model_epoch_10.ckpt

[INFO] epoch/step=11/1100000 | loss=-5.19385 | ploss=-5.18872 | vloss=4446152.97982 | entropy=-5.12614 | reward=48.02662

[INFO] epoch/step=11/1150000 | loss=-5.19082 | ploss=-5.18569 | vloss=5036552.01541 | entropy=-5.12621 | reward=53.99128

[INFO] epoch/step=11/1200000 | loss=-5.18234 | ploss=-5.17721 | vloss=3990832.09813 | entropy=-5.12967 | reward=43.52876

[INFO] Save model to ./tmp/Amazon_Beauty/train_RL_agent/checkpoint/policy_model_epoch_11.ckpt

[INFO] epoch/step=12/1250000 | loss=-5.19156 | ploss=-5.18643 | vloss=4316152.48033 | entropy=-5.13027 | reward=46.65616

[INFO] epoch/step=12/1300000 | loss=-5.18371 | ploss=-5.17859 | vloss=4314506.65732 | entropy=-5.12485 | reward=46.88915

[INFO] Save model to ./tmp/Amazon_Beauty/train_RL_agent/checkpoint/policy_model_epoch_12.ckpt

[INFO] epoch/step=13/1350000 | loss=-5.18989 | ploss=-5.18477 | vloss=4602621.70245 | entropy=-5.12734 | reward=49.51088

[INFO] epoch/step=13/1400000 | loss=-5.20595 | ploss=-5.20082 | vloss=5054650.91865 | entropy=-5.12984 | reward=54.06065

[INFO] epoch/step=13/1450000 | loss=-5.19581 | ploss=-5.19068 | vloss=3879457.50876 | entropy=-5.13066 | reward=42.51674

[INFO] Save model to ./tmp/Amazon_Beauty/train_RL_agent/checkpoint/policy_model_epoch_13.ckpt
[INFO] epoch/step=14/1500000 | loss=-5.19466 | ploss=-5.18954 | vloss=4406603.86778 | entropy=-5.12660 | reward=47.42092
[INFO] epoch/step=14/1550000 | loss=-5.20138 | ploss=-5.19625 | vloss=4697503.58773 | entropy=-5.12954 | reward=50.58073
[INFO] Save model to ./tmp/Amazon_Beauty/train_RL_agent/checkpoint/policy_model_epoch_14.ckpt
[INFO] epoch/step=15/1600000 | loss=-5.19240 | ploss=-5.18727 | vloss=4466480.74994 | entropy=-5.12637 | reward=48.32061
[INFO] epoch/step=15/1650000 | loss=-5.18748 | ploss=-5.18235 | vloss=3946682.98220 | entropy=-5.12847 | reward=43.08671
[INFO] epoch/step=15/1700000 | loss=-5.20932 | ploss=-5.20419 | vloss=4533721.91125 | entropy=-5.12922 | reward=48.92976
[INFO] Save model to ./tmp/Amazon_Beauty/train_RL_agent/checkpoint/policy_model_epoch_15.ckpt
[INFO] epoch/step=16/1750000 | loss=-5.21791 | ploss=-5.21278 | vloss=4845011.11958 | entropy=-5.12693 | reward=52.07321
[INFO] epoch/step=16/1800000 | loss=-5.19706 | ploss=-5.19193 | vloss=4389647.21615 | entropy=-5.12873 | reward=47.45688
[INFO] Save model to ./tmp/Amazon_Beauty/train_RL_agent/checkpoint/policy_model_epoch_16.ckpt
[INFO] epoch/step=17/1850000 | loss=-5.21043 | ploss=-5.20530 | vloss=4082065.90157 | entropy=-5.12965 | reward=44.35486
[INFO] epoch/step=17/1900000 | loss=-5.19964 | ploss=-5.19451 | vloss=4340735.09724 | entropy=-5.12872 | reward=47.08672
[INFO] epoch/step=17/1950000 | loss=-5.21898 | ploss=-5.21385 | vloss=4535933.71734 | entropy=-5.12678 | reward=48.99909
[INFO] Save model to ./tmp/Amazon_Beauty/train_RL_agent/checkpoint/policy_model_epoch_17.ckpt
[INFO] epoch/step=18/2000000 | loss=-5.21083 | ploss=-5.20570 | vloss=4347305.80413 | entropy=-5.13014 | reward=47.13424
[INFO] epoch/step=18/2050000 | loss=-5.20363 | ploss=-5.19850 | vloss=4457709.54212 | entropy=-5.12910 | reward=48.11247
[INFO] Save model to ./tmp/Amazon_Beauty/train_RL_agent/checkpoint/policy_model_epoch_18.ckpt
[INFO] epoch/step=19/2100000 | loss=-5.21721 | ploss=-5.21208 | vloss=4494552.17904 | entropy=-5.12953 | reward=48.45997
[INFO] epoch/step=19/2150000 | loss=-5.22136 | ploss=-5.21623 | vloss=4306457.72722 | entropy=-5.12885 | reward=46.64664
[INFO] Save model to ./tmp/Amazon_Beauty/train_RL_agent/checkpoint/policy_model_epoch_19.ckpt
[INFO] epoch/step=20/2200000 | loss=-5.21389 | ploss=-5.20876 | vloss=4637651.77414 | entropy=-5.12443 | reward=49.98204
[INFO] epoch/step=20/2250000 | loss=-5.22084 | ploss=-5.21571 | vloss=4463662.10133 | entropy=-5.12946 | reward=48.21789
[INFO] epoch/step=20/2300000 | loss=-5.21115 | ploss=-5.20602 | vloss=4218019.42992 | entropy=-5.13032 | reward=45.84278
[INFO] Save model to ./tmp/Amazon_Beauty/train_RL_agent/checkpoint/policy_model_epoch_20.ckpt
[INFO] epoch/step=21/2350000 | loss=-5.22271 | ploss=-5.21758 | vloss=4725592.92454 | entropy=-5.12804 | reward=50.77191
[INFO] epoch/step=21/2400000 | loss=-5.20831 | ploss=-5.20318 | vloss=3925023.55737 | entropy=-5.12749 | reward=42.89354
[INFO] Save model to ./tmp/Amazon_Beauty/train_RL_agent/checkpoint/policy_model_epoch_21.ckpt
[INFO] epoch/step=22/2450000 | loss=-5.22775 | ploss=-5.22262 | vloss=4717465.42719 | entropy=-5.12967 | reward=50.77919
[INFO] epoch/step=22/2500000 | loss=-5.21859 | ploss=-5.21346 | vloss=4497186.49104 | entropy=-5.12668 | reward=48.64155
[INFO] epoch/step=22/2550000 | loss=-5.21945 | ploss=-5.21432 | vloss=4582170.40213 | entropy=-5.12901 | reward=49.30029
[INFO] Save model to ./tmp/Amazon_Beauty/train_RL_agent/checkpoint/policy_model_epoch_22.ckpt
[INFO] epoch/step=23/2600000 | loss=-5.22704 | ploss=-5.22191 | vloss=4360436.85146 | entropy=-5.12733 | reward=47.28542
[INFO] epoch/step=23/2650000 | loss=-5.22406 | ploss=-5.21893 | vloss=4147816.80789 | entropy=-5.13012 | reward=45.08117

1331 [INFO] Save model to ./tmp/Amazon_Beauty/train_RL_agent/checkpoint/policy_model_epoch_23.ckpt
1332 [INFO] epoch/step=24/2700000 | loss=-5.22711 | ploss=-5.22199 | vloss=4639652.04923 | entropy=-
1333 5.12701 | reward=49.91526
1334 [INFO] epoch/step=24/2750000 | loss=-5.21991 | ploss=-5.21478 | vloss=3843942.30477 | entropy=-
1335 5.12858 | reward=42.07761
1336 [INFO] epoch/step=24/2800000 | loss=-5.23178 | ploss=-5.22665 | vloss=4761652.43630 | entropy=-
1337 5.12718 | reward=51.18810
1338 [INFO] Save model to ./tmp/Amazon_Beauty/train_RL_agent/checkpoint/policy_model_epoch_24.ckpt
1339 [INFO] epoch/step=25/2850000 | loss=-5.23232 | ploss=-5.22719 | vloss=4434564.37252 | entropy=-
1340 5.13032 | reward=47.65593
1341 [INFO] epoch/step=25/2900000 | loss=-5.22644 | ploss=-5.22132 | vloss=4522836.96352 | entropy=-
1342 5.12828 | reward=48.90171
1343 [INFO] Save model to ./tmp/Amazon_Beauty/train_RL_agent/checkpoint/policy_model_epoch_25.ckpt
1344 [INFO] epoch/step=26/2950000 | loss=-5.23346 | ploss=-5.22833 | vloss=4574740.16211 | entropy=-
1345 5.12909 | reward=49.29226
1346 [INFO] epoch/step=26/3000000 | loss=-5.22967 | ploss=-5.22454 | vloss=4383210.14246 | entropy=-
1347 5.12779 | reward=47.41295
1348 [INFO] epoch/step=26/3050000 | loss=-5.24471 | ploss=-5.23958 | vloss=4416845.19831 | entropy=-
1349 5.12961 | reward=47.84085
1350 [INFO] Save model to ./tmp/Amazon_Beauty/train_RL_agent/checkpoint/policy_model_epoch_26.ckpt
1351 [INFO] epoch/step=27/3100000 | loss=-5.24308 | ploss=-5.23795 | vloss=4765501.20468 | entropy=-
1352 5.12824 | reward=51.12928
1353 [INFO] epoch/step=27/3150000 | loss=-5.25402 | ploss=-5.24889 | vloss=4252540.47243 | entropy=-
1354 5.12991 | reward=46.16449
1355 [INFO] Save model to ./tmp/Amazon_Beauty/train_RL_agent/checkpoint/policy_model_epoch_27.ckpt
1356 [INFO] epoch/step=28/3200000 | loss=-5.23712 | ploss=-5.23199 | vloss=4504370.29838 | entropy=-
1357 5.12717 | reward=48.60557
1358 [INFO] epoch/step=28/3250000 | loss=-5.24461 | ploss=-5.23948 | vloss=4876599.79624 | entropy=-
1359 5.12718 | reward=52.31020
1360 [INFO] epoch/step=28/3300000 | loss=-5.22768 | ploss=-5.22255 | vloss=3870988.79385 | entropy=-
1361 5.12856 | reward=42.46147
1362 [INFO] Save model to ./tmp/Amazon_Beauty/train_RL_agent/checkpoint/policy_model_epoch_28.ckpt
1363 [INFO] epoch/step=29/3350000 | loss=-5.24413 | ploss=-5.23900 | vloss=4270800.28440 | entropy=-
1364 5.12982 | reward=46.30982
1365 [INFO] epoch/step=29/3400000 | loss=-5.23348 | ploss=-5.22835 | vloss=4773626.82109 | entropy=-
1366 5.12398 | reward=51.33253
1367 [INFO] Save model to ./tmp/Amazon_Beauty/train_RL_agent/checkpoint/policy_model_epoch_29.ckpt
1368 [INFO] epoch/step=30/3450000 | loss=-5.24043 | ploss=-5.23530 | vloss=4467858.16428 | entropy=-
1369 5.13192 | reward=48.19559
1370 [INFO] epoch/step=30/3500000 | loss=-5.26002 | ploss=-5.25490 | vloss=4217143.27471 | entropy=-
1371 5.12787 | reward=45.86561
1372 [INFO] Save model to ./tmp/Amazon_Beauty/train_RL_agent/checkpoint/policy_model_epoch_30.ckpt
1373 [INFO] epoch/step=31/3550000 | loss=-5.24365 | ploss=-5.23853 | vloss=4537328.29491 | entropy=-
1374 5.12908 | reward=48.82095
1375 [INFO] epoch/step=31/3600000 | loss=-5.24601 | ploss=-5.24088 | vloss=4267593.62261 | entropy=-
1376 5.12765 | reward=46.20485
1377 [INFO] epoch/step=31/3650000 | loss=-5.25807 | ploss=-5.25294 | vloss=4472713.43733 | entropy=-
1378 5.12910 | reward=48.43162
1379 [INFO] Save model to ./tmp/Amazon_Beauty/train_RL_agent/checkpoint/policy_model_epoch_31.ckpt
1380 [INFO] epoch/step=32/3700000 | loss=-5.25107 | ploss=-5.24594 | vloss=4354556.52781 | entropy=-
1381 5.12725 | reward=47.13843
1382 [INFO] epoch/step=32/3750000 | loss=-5.25077 | ploss=-5.24564 | vloss=4975833.71739 | entropy=-
1383 5.12837 | reward=53.20915
1384 [INFO] Save model to ./tmp/Amazon_Beauty/train_RL_agent/checkpoint/policy_model_epoch_32.ckpt
1385 [INFO] epoch/step=33/3800000 | loss=-5.25814 | ploss=-5.25301 | vloss=4089229.82385 | entropy=-
1386 5.12971 | reward=44.64225
1387 [INFO] epoch/step=33/3850000 | loss=-5.24438 | ploss=-5.23925 | vloss=4042416.13571 | entropy=-
1388 5.12908 | reward=44.06524

1389 [INFO] epoch/step=33/3900000 | loss=-5.25373 | ploss=-5.24860 | vloss=4821518.14169 | entropy=-
1390 5.12831 | reward=51.67801
1391 [INFO] Save model to ./tmp/Amazon_Beauty/train_RL_agent/checkpoint/policy_model_epoch_33.ckpt
1392 [INFO] epoch/step=34/3950000 | loss=-5.25583 | ploss=-5.25070 | vloss=4214193.30262 | entropy=-
1393 5.12696 | reward=45.82688
1394 [INFO] epoch/step=34/4000000 | loss=-5.25015 | ploss=-5.24502 | vloss=4245857.27149 | entropy=-
1395 5.12936 | reward=46.10695
1396 [INFO] Save model to ./tmp/Amazon_Beauty/train_RL_agent/checkpoint/policy_model_epoch_34.ckpt
1397 [INFO] epoch/step=35/4050000 | loss=-5.27522 | ploss=-5.27009 | vloss=4768032.38682 | entropy=-
1398 5.12843 | reward=51.18764
1399 [INFO] epoch/step=35/4100000 | loss=-5.26051 | ploss=-5.25538 | vloss=4418224.77573 | entropy=-
1400 5.12792 | reward=47.89293
1401 [INFO] epoch/step=35/4150000 | loss=-5.25635 | ploss=-5.25122 | vloss=4368803.56532 | entropy=-
1402 5.12761 | reward=47.26776
1403 [INFO] Save model to ./tmp/Amazon_Beauty/train_RL_agent/checkpoint/policy_model_epoch_35.ckpt
1404 [INFO] epoch/step=36/4200000 | loss=-5.26278 | ploss=-5.25765 | vloss=3948211.96564 | entropy=-
1405 5.12990 | reward=43.00695
1406 [INFO] epoch/step=36/4250000 | loss=-5.27052 | ploss=-5.26539 | vloss=4890526.06959 | entropy=-
1407 5.12659 | reward=52.61912
1408 [INFO] Save model to ./tmp/Amazon_Beauty/train_RL_agent/checkpoint/policy_model_epoch_36.ckpt
1409 [INFO] epoch/step=37/4300000 | loss=-5.25966 | ploss=-5.25453 | vloss=4544468.61695 | entropy=-
1410 5.12798 | reward=48.83717
1411 [INFO] epoch/step=37/4350000 | loss=-5.26344 | ploss=-5.25832 | vloss=4574914.65858 | entropy=-
1412 5.12620 | reward=49.35491
1413 [INFO] epoch/step=37/4400000 | loss=-5.27333 | ploss=-5.26820 | vloss=4334676.88551 | entropy=-
1414 5.12999 | reward=46.97091
1415 [INFO] Save model to ./tmp/Amazon_Beauty/train_RL_agent/checkpoint/policy_model_epoch_37.ckpt
1416 [INFO] epoch/step=38/4450000 | loss=-5.26319 | ploss=-5.25806 | vloss=3966504.60229 | entropy=-
1417 5.12887 | reward=43.21221
1418 [INFO] epoch/step=38/4500000 | loss=-5.26960 | ploss=-5.26448 | vloss=4852719.89353 | entropy=-
1419 5.12646 | reward=52.06486
1420 [INFO] Save model to ./tmp/Amazon_Beauty/train_RL_agent/checkpoint/policy_model_epoch_38.ckpt
1421 [INFO] epoch/step=39/4550000 | loss=-5.27134 | ploss=-5.26621 | vloss=4081985.42478 | entropy=-
1422 5.12919 | reward=44.58272
1423 [INFO] epoch/step=39/4600000 | loss=-5.27423 | ploss=-5.26910 | vloss=4607527.42205 | entropy=-
1424 5.12953 | reward=49.65468
1425 [INFO] Save model to ./tmp/Amazon_Beauty/train_RL_agent/checkpoint/policy_model_epoch_39.ckpt
1426 [INFO] epoch/step=40/4650000 | loss=-5.27584 | ploss=-5.27071 | vloss=4570618.79734 | entropy=-
1427 5.12992 | reward=49.22854
1428 [INFO] epoch/step=40/4700000 | loss=-5.27019 | ploss=-5.26507 | vloss=4464816.00219 | entropy=-
1429 5.12487 | reward=48.29851
1430 [INFO] epoch/step=40/4750000 | loss=-5.28266 | ploss=-5.27753 | vloss=4774953.90760 | entropy=-
1431 5.13182 | reward=51.28884
1432 [INFO] Save model to ./tmp/Amazon_Beauty/train_RL_agent/checkpoint/policy_model_epoch_40.ckpt
1433 [INFO] epoch/step=41/4800000 | loss=-5.27955 | ploss=-5.27442 | vloss=4146638.57956 | entropy=-
1434 5.12745 | reward=44.96761
1435 [INFO] epoch/step=41/4850000 | loss=-5.27777 | ploss=-5.27264 | vloss=4341878.45171 | entropy=-
1436 5.12950 | reward=47.00719
1437 [INFO] Save model to ./tmp/Amazon_Beauty/train_RL_agent/checkpoint/policy_model_epoch_41.ckpt
1438 [INFO] epoch/step=42/4900000 | loss=-5.25693 | ploss=-5.25180 | vloss=4341769.22033 | entropy=-
1439 5.12932 | reward=47.08022
1440 [INFO] epoch/step=42/4950000 | loss=-5.27414 | ploss=-5.26901 | vloss=4600189.17547 | entropy=-
1441 5.13106 | reward=49.45321
1442 [INFO] epoch/step=42/5000000 | loss=-5.28250 | ploss=-5.27738 | vloss=4199458.33428 | entropy=-
1443 5.12650 | reward=45.74931
1444 [INFO] Save model to ./tmp/Amazon_Beauty/train_RL_agent/checkpoint/policy_model_epoch_42.ckpt
1445 [INFO] epoch/step=43/5050000 | loss=-5.29504 | ploss=-5.28991 | vloss=4740016.42597 | entropy=-
1446 5.12757 | reward=50.92592

1447 [INFO] epoch/step=43/5100000 | loss=-5.28239 | ploss=-5.27726 | vloss=4686084.99812 | entropy=-
1448 5.12641 | reward=50.53399
1449 [INFO] Save model to ./tmp/Amazon_Beauty/train_RL_agent/checkpoint/policy_model_epoch_43.ckpt
1450 [INFO] epoch/step=44/5150000 | loss=-5.26906 | ploss=-5.26393 | vloss=4035515.86919 | entropy=-
1451 5.12942 | reward=43.80434
1452 [INFO] epoch/step=44/5200000 | loss=-5.29025 | ploss=-5.28512 | vloss=4237456.12681 | entropy=-
1453 5.12916 | reward=46.08522
1454 [INFO] epoch/step=44/5250000 | loss=-5.27845 | ploss=-5.27332 | vloss=4638738.19628 | entropy=-
1455 5.12791 | reward=50.01882
1456 [INFO] Save model to ./tmp/Amazon_Beauty/train_RL_agent/checkpoint/policy_model_epoch_44.ckpt
1457 [INFO] epoch/step=45/5300000 | loss=-5.28086 | ploss=-5.27573 | vloss=4192714.67664 | entropy=-
1458 5.12981 | reward=45.40344
1459 [INFO] epoch/step=45/5350000 | loss=-5.29039 | ploss=-5.28527 | vloss=4600612.08954 | entropy=-
1460 5.12647 | reward=49.64592
1461 [INFO] Save model to ./tmp/Amazon_Beauty/train_RL_agent/checkpoint/policy_model_epoch_45.ckpt
1462 [INFO] epoch/step=46/5400000 | loss=-5.29398 | ploss=-5.28885 | vloss=4764986.64040 | entropy=-
1463 5.12864 | reward=51.20260
1464 [INFO] epoch/step=46/5450000 | loss=-5.27565 | ploss=-5.27052 | vloss=4283215.48872 | entropy=-
1465 5.12969 | reward=46.32399
1466 [INFO] epoch/step=46/5500000 | loss=-5.29662 | ploss=-5.29149 | vloss=4437393.67055 | entropy=-
1467 5.12422 | reward=48.15583
1468 [INFO] Save model to ./tmp/Amazon_Beauty/train_RL_agent/checkpoint/policy_model_epoch_46.ckpt
1469 [INFO] epoch/step=47/5550000 | loss=-5.27647 | ploss=-5.27134 | vloss=4678358.39884 | entropy=-
1470 5.12597 | reward=50.39202
1471 [INFO] epoch/step=47/5600000 | loss=-5.28752 | ploss=-5.28239 | vloss=4290074.56897 | entropy=-
1472 5.13135 | reward=46.47433
1473 [INFO] Save model to ./tmp/Amazon_Beauty/train_RL_agent/checkpoint/policy_model_epoch_47.ckpt
1474 [INFO] epoch/step=48/5650000 | loss=-5.28266 | ploss=-5.27753 | vloss=3840302.32574 | entropy=-
1475 5.12785 | reward=42.06977
1476 [INFO] epoch/step=48/5700000 | loss=-5.29223 | ploss=-5.28710 | vloss=4512158.82379 | entropy=-
1477 5.12850 | reward=48.77678
1478 [INFO] Save model to ./tmp/Amazon_Beauty/train_RL_agent/checkpoint/policy_model_epoch_48.ckpt
1479 [INFO] epoch/step=49/5750000 | loss=-5.29482 | ploss=-5.28969 | vloss=4720175.96027 | entropy=-
1480 5.12814 | reward=50.67476
1481 [INFO] epoch/step=49/5800000 | loss=-5.29192 | ploss=-5.28679 | vloss=4795095.46180 | entropy=-
1482 5.12947 | reward=51.49693
1483 [INFO] epoch/step=49/5850000 | loss=-5.31035 | ploss=-5.30523 | vloss=4567261.45881 | entropy=-
1484 5.12718 | reward=49.27707
1485 [INFO] Save model to ./tmp/Amazon_Beauty/train_RL_agent/checkpoint/policy_model_epoch_49.ckpt
1486 [INFO] epoch/step=50/5900000 | loss=-5.29735 | ploss=-5.29222 | vloss=4202066.95943 | entropy=-
1487 5.12788 | reward=45.64420
1488 [INFO] epoch/step=50/5950000 | loss=-5.30838 | ploss=-5.30325 | vloss=4587423.28417 | entropy=-
1489 5.12742 | reward=49.48744
1490 [INFO] Save model to ./tmp/Amazon_Beauty/train_RL_agent/checkpoint/policy_model_epoch_50.ckpt
1491 [INFO] epoch/step=51/6000000 | loss=-5.28347 | ploss=-5.27835 | vloss=4151272.98279 | entropy=-
1492 5.12894 | reward=45.06981
1493 [INFO] epoch/step=51/6050000 | loss=-5.28986 | ploss=-5.28473 | vloss=4360621.73513 | entropy=-
1494 5.12685 | reward=47.20722
1495 [INFO] epoch/step=51/6100000 | loss=-5.29936 | ploss=-5.29424 | vloss=4670561.48589 | entropy=-
1496 5.12870 | reward=50.24235
1497 [INFO] Save model to ./tmp/Amazon_Beauty/train_RL_agent/checkpoint/policy_model_epoch_51.ckpt
1498 [INFO] epoch/step=52/6150000 | loss=-5.30981 | ploss=-5.30468 | vloss=4709930.30151 | entropy=-
1499 5.12971 | reward=50.65128
1500 [INFO] epoch/step=52/6200000 | loss=-5.29067 | ploss=-5.28555 | vloss=4139243.97294 | entropy=-
1501 5.12536 | reward=45.10017
1502 [INFO] Save model to ./tmp/Amazon_Beauty/train_RL_agent/checkpoint/policy_model_epoch_52.ckpt
1503 [INFO] epoch/step=53/6250000 | loss=-5.30197 | ploss=-5.29685 | vloss=4332273.84169 | entropy=-
1504 5.12764 | reward=46.84077

1505 [INFO] epoch/step=53/6300000 | loss=-5.31531 | ploss=-5.31018 | vloss=4181135.74232 | entropy=-
1506 5.13126 | reward=45.47048
1507 [INFO] epoch/step=53/6350000 | loss=-5.30689 | ploss=-5.30177 | vloss=4694765.41586 | entropy=-
1508 5.12740 | reward=50.51527
1509 [INFO] Save model to ./tmp/Amazon_Beauty/train_RL_agent/checkpoint/policy_model_epoch_53.ckpt
1510 [INFO] epoch/step=54/6400000 | loss=-5.30910 | ploss=-5.30397 | vloss=4270157.81386 | entropy=-
1511 5.13129 | reward=46.17421
1512 [INFO] epoch/step=54/6450000 | loss=-5.30511 | ploss=-5.29998 | vloss=4677363.16943 | entropy=-
1513 5.12740 | reward=50.36397
1514 [INFO] Save model to ./tmp/Amazon_Beauty/train_RL_agent/checkpoint/policy_model_epoch_54.ckpt
1515 [INFO] epoch/step=55/6500000 | loss=-5.30805 | ploss=-5.30293 | vloss=4323329.41066 | entropy=-
1516 5.12781 | reward=46.89741
1517 [INFO] epoch/step=55/6550000 | loss=-5.31824 | ploss=-5.31311 | vloss=4559698.98236 | entropy=-
1518 5.12774 | reward=49.28191
1519 [INFO] epoch/step=55/6600000 | loss=-5.30916 | ploss=-5.30403 | vloss=4413714.62534 | entropy=-
1520 5.12826 | reward=47.63709
1521 [INFO] Save model to ./tmp/Amazon_Beauty/train_RL_agent/checkpoint/policy_model_epoch_55.ckpt
1522 [INFO] epoch/step=56/6650000 | loss=-5.30714 | ploss=-5.30201 | vloss=4448223.91749 | entropy=-
1523 5.12895 | reward=47.99492
1524 [INFO] epoch/step=56/6700000 | loss=-5.31131 | ploss=-5.30618 | vloss=4036117.63613 | entropy=-
1525 5.13063 | reward=44.04577
1526 [INFO] Save model to ./tmp/Amazon_Beauty/train_RL_agent/checkpoint/policy_model_epoch_56.ckpt
1527 [INFO] epoch/step=57/6750000 | loss=-5.31354 | ploss=-5.30841 | vloss=4438076.27417 | entropy=-
1528 5.12763 | reward=48.02351
1529 [INFO] epoch/step=57/6800000 | loss=-5.32005 | ploss=-5.31492 | vloss=4521661.90331 | entropy=-
1530 5.12796 | reward=48.81187
1531 [INFO] epoch/step=57/6850000 | loss=-5.32354 | ploss=-5.31841 | vloss=4593894.23209 | entropy=-
1532 5.12987 | reward=49.47737
1533 [INFO] Save model to ./tmp/Amazon_Beauty/train_RL_agent/checkpoint/policy_model_epoch_57.ckpt
1534 [INFO] epoch/step=58/6900000 | loss=-5.32001 | ploss=-5.31488 | vloss=4537419.24888 | entropy=-
1535 5.13091 | reward=48.92538
1536 [INFO] epoch/step=58/6950000 | loss=-5.31703 | ploss=-5.31191 | vloss=4229908.81615 | entropy=-
1537 5.12782 | reward=45.91667
1538 [INFO] Save model to ./tmp/Amazon_Beauty/train_RL_agent/checkpoint/policy_model_epoch_58.ckpt
1539 [INFO] epoch/step=59/7000000 | loss=-5.30947 | ploss=-5.30434 | vloss=4394131.06442 | entropy=-
1540 5.12515 | reward=47.55128
1541 [INFO] epoch/step=59/7050000 | loss=-5.31740 | ploss=-5.31227 | vloss=4502068.65970 | entropy=-
1542 5.12803 | reward=48.73302
1543 [INFO] Save model to ./tmp/Amazon_Beauty/train_RL_agent/checkpoint/policy_model_epoch_59.ckpt
1544 [INFO] epoch/step=60/7100000 | loss=-5.32192 | ploss=-5.31679 | vloss=4381954.77226 | entropy=-
1545 5.12866 | reward=47.27031
1546 [INFO] epoch/step=60/7150000 | loss=-5.31044 | ploss=-5.30531 | vloss=4088884.41093 | entropy=-
1547 5.13071 | reward=44.39230
1548 [INFO] epoch/step=60/7200000 | loss=-5.32494 | ploss=-5.31981 | vloss=4809553.06763 | entropy=-
1549 5.12764 | reward=51.70765
1550 [INFO] Save model to ./tmp/Amazon_Beauty/train_RL_agent/checkpoint/policy_model_epoch_60.ckpt
1551 [INFO] epoch/step=61/7250000 | loss=-5.31976 | ploss=-5.31463 | vloss=4408753.64055 | entropy=-
1552 5.12873 | reward=47.52556
1553 [INFO] epoch/step=61/7300000 | loss=-5.32353 | ploss=-5.31840 | vloss=4444895.74259 | entropy=-
1554 5.12654 | reward=48.04435
1555 [INFO] Save model to ./tmp/Amazon_Beauty/train_RL_agent/checkpoint/policy_model_epoch_61.ckpt
1556 [INFO] epoch/step=62/7350000 | loss=-5.32817 | ploss=-5.32304 | vloss=4658570.17316 | entropy=-
1557 5.12899 | reward=50.13583
1558 [INFO] epoch/step=62/7400000 | loss=-5.31767 | ploss=-5.31254 | vloss=4017377.57903 | entropy=-
1559 5.12677 | reward=43.95642
1560 [INFO] epoch/step=62/7450000 | loss=-5.32859 | ploss=-5.32346 | vloss=4806390.33207 | entropy=-
1561 5.13065 | reward=51.45823
1562 [INFO] Save model to ./tmp/Amazon_Beauty/train_RL_agent/checkpoint/policy_model_epoch_62.ckpt

[INFO] epoch/step=63/7500000 | loss=-5.33910 | ploss=-5.33397 | vloss=4172299.49900 | entropy=-5.12829 | reward=45.40763

[INFO] epoch/step=63/7550000 | loss=-5.32386 | ploss=-5.31873 | vloss=4884399.75102 | entropy=-5.12794 | reward=52.42910

[INFO] Save model to ./tmp/Amazon_Beauty/train_RL_agent/checkpoint/policy_model_epoch_63.ckpt

[INFO] epoch/step=64/7600000 | loss=-5.31876 | ploss=-5.31363 | vloss=4328774.62701 | entropy=-5.13144 | reward=46.77559

[INFO] epoch/step=64/7650000 | loss=-5.33150 | ploss=-5.32637 | vloss=4658958.85717 | entropy=-5.12854 | reward=50.17498

[INFO] epoch/step=64/7700000 | loss=-5.31706 | ploss=-5.31193 | vloss=4100689.79614 | entropy=-5.12804 | reward=44.65526

[INFO] Save model to ./tmp/Amazon_Beauty/train_RL_agent/checkpoint/policy_model_epoch_64.ckpt

[INFO] epoch/step=65/7750000 | loss=-5.34310 | ploss=-5.33797 | vloss=4333376.30340 | entropy=-5.12793 | reward=47.00684

[INFO] epoch/step=65/7800000 | loss=-5.33078 | ploss=-5.32566 | vloss=4655115.68514 | entropy=-5.12761 | reward=50.05611

[INFO] Save model to ./tmp/Amazon_Beauty/train_RL_agent/checkpoint/policy_model_epoch_65.ckpt

[INFO] epoch/step=66/7850000 | loss=-5.32639 | ploss=-5.32127 | vloss=4103602.24999 | entropy=-5.12656 | reward=44.60437

[INFO] epoch/step=66/7900000 | loss=-5.32829 | ploss=-5.32316 | vloss=4861050.31097 | entropy=-5.13048 | reward=52.13355

[INFO] epoch/step=66/7950000 | loss=-5.33622 | ploss=-5.33109 | vloss=4276722.97300 | entropy=-5.12865 | reward=46.44858

[INFO] Save model to ./tmp/Amazon_Beauty/train_RL_agent/checkpoint/policy_model_epoch_66.ckpt

[INFO] epoch/step=67/8000000 | loss=-5.34457 | ploss=-5.33944 | vloss=4413333.94290 | entropy=-5.13069 | reward=47.66827

[INFO] epoch/step=67/8050000 | loss=-5.33190 | ploss=-5.32678 | vloss=4145652.46080 | entropy=-5.12510 | reward=45.13413

[INFO] Save model to ./tmp/Amazon_Beauty/train_RL_agent/checkpoint/policy_model_epoch_67.ckpt

[INFO] epoch/step=68/8100000 | loss=-5.33365 | ploss=-5.32852 | vloss=4680063.02568 | entropy=-5.12711 | reward=50.46790

[INFO] epoch/step=68/8150000 | loss=-5.35050 | ploss=-5.34537 | vloss=4830687.85563 | entropy=-5.13151 | reward=51.86325

[INFO] Save model to ./tmp/Amazon_Beauty/train_RL_agent/checkpoint/policy_model_epoch_68.ckpt

[INFO] epoch/step=69/8200000 | loss=-5.32779 | ploss=-5.32266 | vloss=3986434.47911 | entropy=-5.12819 | reward=43.37727

[INFO] epoch/step=69/8250000 | loss=-5.33734 | ploss=-5.33221 | vloss=4593068.65884 | entropy=-5.12904 | reward=49.49016

[INFO] epoch/step=69/8300000 | loss=-5.34320 | ploss=-5.33807 | vloss=4581220.86193 | entropy=-5.12605 | reward=49.37189

[INFO] Save model to ./tmp/Amazon_Beauty/train_RL_agent/checkpoint/policy_model_epoch_69.ckpt

[INFO] epoch/step=70/8350000 | loss=-5.34012 | ploss=-5.33499 | vloss=4089623.39729 | entropy=-5.12737 | reward=44.72647

[INFO] epoch/step=70/8400000 | loss=-5.34722 | ploss=-5.34209 | vloss=4115744.42589 | entropy=-5.13183 | reward=44.72295

[INFO] Save model to ./tmp/Amazon_Beauty/train_RL_agent/checkpoint/policy_model_epoch_70.ckpt

[INFO] epoch/step=71/8450000 | loss=-5.34891 | ploss=-5.34379 | vloss=4877953.79642 | entropy=-5.12655 | reward=52.07156

[INFO] epoch/step=71/8500000 | loss=-5.34927 | ploss=-5.34414 | vloss=4529071.78933 | entropy=-5.12820 | reward=48.85986

[INFO] epoch/step=71/8550000 | loss=-5.34142 | ploss=-5.33629 | vloss=4419601.27354 | entropy=-5.12908 | reward=47.87687

[INFO] Save model to ./tmp/Amazon_Beauty/train_RL_agent/checkpoint/policy_model_epoch_71.ckpt

[INFO] epoch/step=72/8600000 | loss=-5.34381 | ploss=-5.33868 | vloss=4638059.48820 | entropy=-5.12648 | reward=49.88984

[INFO] epoch/step=72/8650000 | loss=-5.32583 | ploss=-5.32070 | vloss=3972702.11506 | entropy=-5.12843 | reward=43.40311

[INFO] Save model to ./tmp/Amazon_Beauty/train_RL_agent/checkpoint/policy_model_epoch_72.ckpt

[INFO] epoch/step=73/8700000 | loss=-5.34890 | ploss=-5.34377 | vloss=4409789.97848 | entropy=-5.12880 | reward=47.62893

[INFO] epoch/step=73/8750000 | loss=-5.34711 | ploss=-5.34199 | vloss=4291236.35068 | entropy=-5.12782 | reward=46.48619

[INFO] epoch/step=73/8800000 | loss=-5.35308 | ploss=-5.34795 | vloss=4598590.82528 | entropy=-5.12585 | reward=49.67223

[INFO] Save model to ./tmp/Amazon_Beauty/train_RL_agent/checkpoint/policy_model_epoch_73.ckpt

[INFO] epoch/step=74/8850000 | loss=-5.35571 | ploss=-5.35058 | vloss=4647427.46820 | entropy=-5.13046 | reward=50.15363

[INFO] epoch/step=74/8900000 | loss=-5.35060 | ploss=-5.34547 | vloss=4303036.28325 | entropy=-5.12785 | reward=46.58256

[INFO] Save model to ./tmp/Amazon_Beauty/train_RL_agent/checkpoint/policy_model_epoch_74.ckpt

[INFO] epoch/step=75/8950000 | loss=-5.34157 | ploss=-5.33644 | vloss=4713700.69608 | entropy=-5.12779 | reward=50.51763

[INFO] epoch/step=75/9000000 | loss=-5.34900 | ploss=-5.34387 | vloss=4714432.58499 | entropy=-5.13039 | reward=50.76700

[INFO] epoch/step=75/9050000 | loss=-5.35295 | ploss=-5.34782 | vloss=4050567.42453 | entropy=-5.12659 | reward=44.15199

[INFO] Save model to ./tmp/Amazon_Beauty/train_RL_agent/checkpoint/policy_model_epoch_75.ckpt

[INFO] epoch/step=76/9100000 | loss=-5.34513 | ploss=-5.34000 | vloss=4224079.64386 | entropy=-5.12840 | reward=45.99916

[INFO] epoch/step=76/9150000 | loss=-5.34719 | ploss=-5.34206 | vloss=4550653.28390 | entropy=-5.12908 | reward=49.01462

[INFO] Save model to ./tmp/Amazon_Beauty/train_RL_agent/checkpoint/policy_model_epoch_76.ckpt

[INFO] epoch/step=77/9200000 | loss=-5.36402 | ploss=-5.35889 | vloss=4317675.51379 | entropy=-5.12819 | reward=46.77411

[INFO] epoch/step=77/9250000 | loss=-5.36185 | ploss=-5.35672 | vloss=4250947.70711 | entropy=-5.12677 | reward=46.14189

[INFO] Save model to ./tmp/Amazon_Beauty/train_RL_agent/checkpoint/policy_model_epoch_77.ckpt

[INFO] epoch/step=78/9300000 | loss=-5.34895 | ploss=-5.34382 | vloss=4665140.05015 | entropy=-5.12954 | reward=50.16302

[INFO] epoch/step=78/9350000 | loss=-5.35143 | ploss=-5.34630 | vloss=4382028.48035 | entropy=-5.12902 | reward=47.44515

[INFO] epoch/step=78/9400000 | loss=-5.34403 | ploss=-5.33890 | vloss=4560901.39102 | entropy=-5.12563 | reward=49.25341

[INFO] Save model to ./tmp/Amazon_Beauty/train_RL_agent/checkpoint/policy_model_epoch_78.ckpt

[INFO] epoch/step=79/9450000 | loss=-5.34980 | ploss=-5.34467 | vloss=4396113.91640 | entropy=-5.12945 | reward=47.44970

[INFO] epoch/step=79/9500000 | loss=-5.36493 | ploss=-5.35980 | vloss=4743393.62287 | entropy=-5.12733 | reward=51.03619

[INFO] Save model to ./tmp/Amazon_Beauty/train_RL_agent/checkpoint/policy_model_epoch_79.ckpt

[INFO] epoch/step=80/9550000 | loss=-5.36692 | ploss=-5.36179 | vloss=4001625.48961 | entropy=-5.13131 | reward=43.67466

[INFO] epoch/step=80/9600000 | loss=-5.36157 | ploss=-5.35644 | vloss=4406957.95941 | entropy=-5.12673 | reward=47.59580

[INFO] epoch/step=80/9650000 | loss=-5.36654 | ploss=-5.36141 | vloss=4554869.18726 | entropy=-5.12958 | reward=49.18934

[INFO] Save model to ./tmp/Amazon_Beauty/train_RL_agent/checkpoint/policy_model_epoch_80.ckpt

[INFO] epoch/step=81/9700000 | loss=-5.36873 | ploss=-5.36360 | vloss=4507765.59382 | entropy=-5.12743 | reward=48.61201

[INFO] epoch/step=81/9750000 | loss=-5.35512 | ploss=-5.34998 | vloss=4371957.16682 | entropy=-5.13162 | reward=47.29370

[INFO] Save model to ./tmp/Amazon_Beauty/train_RL_agent/checkpoint/policy_model_epoch_81.ckpt

[INFO] epoch/step=82/9800000 | loss=-5.35563 | ploss=-5.35050 | vloss=4428990.40534 | entropy=-5.12740 | reward=47.86986

[INFO] epoch/step=82/9850000 | loss=-5.34729 | ploss=-5.34217 | vloss=4882200.92982 | entropy=-5.12360 | reward=52.34699

[INFO] epoch/step=82/9900000 | loss=-5.37296 | ploss=-5.36783 | vloss=4331024.15042 | entropy=-5.12898 | reward=47.02662

1680 [INFO] Save model to ./tmp/Amazon_Beauty/train_RL_agent/checkpoint/policy_model_epoch_82.ckpt
1681 [INFO] epoch/step=83/9950000 | loss=-5.36433 | ploss=-5.35920 | vloss=4137560.44794 | entropy=-
1682 5.12640 | reward=44.97016
1683 [INFO] epoch/step=83/10000000 | loss=-5.36984 | ploss=-5.36471 | vloss=4674159.27769 | entropy=-
1684 5.13057 | reward=50.36802
1685 [INFO] Save model to ./tmp/Amazon_Beauty/train_RL_agent/checkpoint/policy_model_epoch_83.ckpt
1686 [INFO] epoch/step=84/10050000 | loss=-5.37047 | ploss=-5.36534 | vloss=4438180.63938 | entropy=-
1687 5.13091 | reward=47.88905
1688 [INFO] epoch/step=84/10100000 | loss=-5.37340 | ploss=-5.36827 | vloss=4393116.57183 | entropy=-
1689 5.12519 | reward=47.58200
1690 [INFO] epoch/step=84/10150000 | loss=-5.36154 | ploss=-5.35641 | vloss=4354675.80454 | entropy=-
1691 5.13203 | reward=47.11219
1692 [INFO] Save model to ./tmp/Amazon_Beauty/train_RL_agent/checkpoint/policy_model_epoch_84.ckpt
1693 [INFO] epoch/step=85/10200000 | loss=-5.36520 | ploss=-5.36007 | vloss=4084679.71002 | entropy=-
1694 5.12783 | reward=44.44744
1695 [INFO] epoch/step=85/10250000 | loss=-5.36171 | ploss=-5.35658 | vloss=4297894.07855 | entropy=-
1696 5.12655 | reward=46.63234
1697 [INFO] Save model to ./tmp/Amazon_Beauty/train_RL_agent/checkpoint/policy_model_epoch_85.ckpt
1698 [INFO] epoch/step=86/10300000 | loss=-5.37842 | ploss=-5.37329 | vloss=5031599.11354 | entropy=-
1699 5.13042 | reward=53.81258
1700 [INFO] epoch/step=86/10350000 | loss=-5.37481 | ploss=-5.36969 | vloss=4236144.76468 | entropy=-
1701 5.12771 | reward=45.98332
1702 [INFO] epoch/step=86/10400000 | loss=-5.36140 | ploss=-5.35627 | vloss=4390291.32882 | entropy=-
1703 5.12792 | reward=47.56495
1704 [INFO] Save model to ./tmp/Amazon_Beauty/train_RL_agent/checkpoint/policy_model_epoch_86.ckpt
1705 [INFO] epoch/step=87/10450000 | loss=-5.37047 | ploss=-5.36534 | vloss=4883474.20203 | entropy=-
1706 5.12505 | reward=52.43381
1707 [INFO] epoch/step=87/10500000 | loss=-5.37202 | ploss=-5.36689 | vloss=4094504.24685 | entropy=-
1708 5.13090 | reward=44.53121
1709 [INFO] Save model to ./tmp/Amazon_Beauty/train_RL_agent/checkpoint/policy_model_epoch_87.ckpt
1710 [INFO] epoch/step=88/10550000 | loss=-5.37632 | ploss=-5.37119 | vloss=4685580.19669 | entropy=-
1711 5.12942 | reward=50.35640
1712 [INFO] epoch/step=88/10600000 | loss=-5.36828 | ploss=-5.36315 | vloss=4329871.82761 | entropy=-
1713 5.12825 | reward=46.94039
1714 [INFO] Save model to ./tmp/Amazon_Beauty/train_RL_agent/checkpoint/policy_model_epoch_88.ckpt
1715 [INFO] epoch/step=89/10650000 | loss=-5.37325 | ploss=-5.36813 | vloss=4228941.20410 | entropy=-
1716 5.12740 | reward=45.90814
1717 [INFO] epoch/step=89/10700000 | loss=-5.36190 | ploss=-5.35677 | vloss=4299437.03288 | entropy=-
1718 5.12762 | reward=46.61257
1719 [INFO] epoch/step=89/10750000 | loss=-5.38561 | ploss=-5.38048 | vloss=4647984.57886 | entropy=-
1720 5.12859 | reward=50.05633
1721 [INFO] Save model to ./tmp/Amazon_Beauty/train_RL_agent/checkpoint/policy_model_epoch_89.ckpt
1722 [INFO] epoch/step=90/10800000 | loss=-5.37702 | ploss=-5.37189 | vloss=4254399.34713 | entropy=-
1723 5.12879 | reward=46.12443
1724 [INFO] epoch/step=90/10850000 | loss=-5.37162 | ploss=-5.36649 | vloss=4261913.49735 | entropy=-
1725 5.12728 | reward=46.28943
1726 [INFO] Save model to ./tmp/Amazon_Beauty/train_RL_agent/checkpoint/policy_model_epoch_90.ckpt
1727 [INFO] epoch/step=91/10900000 | loss=-5.37393 | ploss=-5.36880 | vloss=4634483.44022 | entropy=-
1728 5.12827 | reward=49.85795
1729 [INFO] epoch/step=91/10950000 | loss=-5.37273 | ploss=-5.36760 | vloss=4700007.29531 | entropy=-
1730 5.13024 | reward=50.66021
1731 [INFO] epoch/step=91/11000000 | loss=-5.38477 | ploss=-5.37964 | vloss=4340515.07168 | entropy=-
1732 5.12599 | reward=47.06956
1733 [INFO] Save model to ./tmp/Amazon_Beauty/train_RL_agent/checkpoint/policy_model_epoch_91.ckpt
1734 [INFO] epoch/step=92/11050000 | loss=-5.36761 | ploss=-5.36248 | vloss=4515673.37385 | entropy=-
1735 5.12796 | reward=48.64116
1736 [INFO] epoch/step=92/11100000 | loss=-5.37917 | ploss=-5.37405 | vloss=4482637.45135 | entropy=-
1737 5.12690 | reward=48.42499

1738   [INFO] Save model to ./tmp/Amazon_Beauty/train_RL_agent/checkpoint/policy_model_epoch_92.ckpt

1739   [INFO] epoch/step=93/11150000 | loss=-5.38155 | ploss=-5.37642 | vloss=4324008.12640 | entropy=-
1740   5.13141 | reward=46.75242

1741   [INFO] epoch/step=93/11200000 | loss=-5.37455 | ploss=-5.36942 | vloss=4078591.36288 | entropy=-
1742   5.13246 | reward=44.38573

1743   [INFO] epoch/step=93/11250000 | loss=-5.38195 | ploss=-5.37683 | vloss=4470164.67443 | entropy=-
1744   5.12573 | reward=48.40018

1745   [INFO] Save model to ./tmp/Amazon_Beauty/train_RL_agent/checkpoint/policy_model_epoch_93.ckpt

1746   [INFO] epoch/step=94/11300000 | loss=-5.38048 | ploss=-5.37536 | vloss=4893848.82396 | entropy=-
1747   5.12672 | reward=52.44811

1748   [INFO] epoch/step=94/11350000 | loss=-5.38217 | ploss=-5.37704 | vloss=4379521.17196 | entropy=-
1749   5.13056 | reward=47.44842

1750   [INFO] Save model to ./tmp/Amazon_Beauty/train_RL_agent/checkpoint/policy_model_epoch_94.ckpt

1751   [INFO] epoch/step=95/11400000 | loss=-5.38820 | ploss=-5.38307 | vloss=4243722.40130 | entropy=-
1752   5.12886 | reward=45.98720

1753   [INFO] epoch/step=95/11450000 | loss=-5.39196 | ploss=-5.38683 | vloss=4380446.15402 | entropy=-
1754   5.12467 | reward=47.47583

1755   [INFO] epoch/step=95/11500000 | loss=-5.38711 | ploss=-5.38198 | vloss=4543044.72892 | entropy=-
1756   5.13080 | reward=48.96635

1757   [INFO] Save model to ./tmp/Amazon_Beauty/train_RL_agent/checkpoint/policy_model_epoch_95.ckpt

1758   [INFO] epoch/step=96/11550000 | loss=-5.38044 | ploss=-5.37531 | vloss=4318065.76435 | entropy=-
1759   5.12709 | reward=46.75008

1760   [INFO] epoch/step=96/11600000 | loss=-5.38714 | ploss=-5.38201 | vloss=4178696.18288 | entropy=-
1761   5.13084 | reward=45.39650

1762   [INFO] Save model to ./tmp/Amazon_Beauty/train_RL_agent/checkpoint/policy_model_epoch_96.ckpt

1763   [INFO] epoch/step=97/11650000 | loss=-5.38397 | ploss=-5.37885 | vloss=4512556.63125 | entropy=-
1764   5.12708 | reward=48.72361

1765   [INFO] epoch/step=97/11700000 | loss=-5.38168 | ploss=-5.37656 | vloss=4015885.42964 | entropy=-
1766   5.12710 | reward=43.89515

1767   [INFO] Save model to ./tmp/Amazon_Beauty/train_RL_agent/checkpoint/policy_model_epoch_97.ckpt

1768   [INFO] epoch/step=98/11750000 | loss=-5.40550 | ploss=-5.40037 | vloss=5076472.73231 | entropy=-
1769   5.12902 | reward=54.19283

1770   [INFO] epoch/step=98/11800000 | loss=-5.39148 | ploss=-5.38634 | vloss=4314604.29173 | entropy=-
1771   5.13028 | reward=46.72921

1772   [INFO] epoch/step=98/11850000 | loss=-5.38347 | ploss=-5.37834 | vloss=4139465.21036 | entropy=-
1773   5.12906 | reward=45.01031

1774   [INFO] Save model to ./tmp/Amazon_Beauty/train_RL_agent/checkpoint/policy_model_epoch_98.ckpt

1775   [INFO] epoch/step=99/11900000 | loss=-5.39627 | ploss=-5.39115 | vloss=5185530.40201 | entropy=-
1776   5.12716 | reward=55.29778

1777   [INFO] epoch/step=99/11950000 | loss=-5.38328 | ploss=-5.37815 | vloss=4109285.71482 | entropy=-
1778   5.12838 | reward=44.72693

1779   [INFO] Save model to ./tmp/Amazon_Beauty/train_RL_agent/checkpoint/policy_model_epoch_99.ckpt

1780   [INFO] epoch/step=100/12000000 | loss=-5.39649 | ploss=-5.39136 | vloss=4342875.08798 | entropy=-
1781   5.12842 | reward=46.97655

1782   [INFO] epoch/step=100/12050000 | loss=-5.40273 | ploss=-5.39760 | vloss=4615036.97909 | entropy=-
1783   5.13101 | reward=49.72319

1784   [INFO] epoch/step=100/12100000 | loss=-5.39149 | ploss=-5.38636 | vloss=4210882.15599 | entropy=-
1785   5.12883 | reward=45.67515

1786   [INFO] Save model to ./tmp/Amazon_Beauty/train_RL_agent/checkpoint/policy_model_epoch_100.ckpt

1787

### A.6.3   Test - RL Agent (MES-CI) - for a user

1789   [INFO]      Namespace(dataset='beauty',      source_name='train_RL_agent',      out-
1790   put_folder='test_RL_agent', users=21001, seed=123, gpu=0, epochs=100, max_acts=250,
1791   max_path_len=3, gamma=0.99, state_history=1, hidden=[512, 256], add_products=False, topk=[10,
1792   10, 12], run_path=True, run_eval=True, debug=True, batch_size=32, is_resume_from_checkpoint=0,
1793   logging_mode='a',      log_file_name='test_agent_log',      checkpoint_folder='checkpoint',
1794   MES_score_option=1, PAS_score_option=1, run_number=1, is_only_run_specific_epoch=1,
1795   device=device(type='cpu'), output_dir='./tmp/Amazon_Beauty/test_RL_agent')

1796 [INFO] model epoch=100 | count (users)=1 | ndcg=100.00000 | recall=100.00000 | hit_ratio=100.00000
1797 | precision=10.00000 | invalid_users=0.00000 | execution_timestamp=2025-05-15 09:33:00.890819
1798 train_labels[uid] : 21001 [11808, 7381, 9141, 10747]
1799 test_labels[uid] : 21001 [11772]
1800 **pred_labels[uid] : 21001 [11772, 8471, 9576, 5351, 1690, 5603, 8015, 1465, 10872, 5934]**
1801 i: j : 21001 0
1802 pred_labels[uid] : 21001 11772
1803 **pred_labels_path[uid] : 21001 (11772, 'user 21001 has purchase product 10747 which was**
1804 **produced_by by brand 201 who produced_by product 11772')**
1805 pred_labels_details[uid] : 21001 (np.float32(3.6954694), np.float32(1.0), np.float32(1.0986123),
1806 np.float32(5693.659), [('self_loop', 'user', 21001), ('purchase', 'product', 10747), ('pro-
1807 duced_by', 'brand', 201), ('produced_by', 'product', 11772)], np.float32(8.231993),
1808 np.float32(-0.4814049), np.float32(0.020631433), np.float32(4231.7305), np.float32(1.8997045),
1809 np.float64(1.6209097319169405), np.float64(2.178499180741751), 6)
1810 pred_probs=1.0 | pred_entropy=1.0986123085021973 | pred_reward=5693.6591796875
1811 | | pred_path=[('self_loop', 'user', 21001), ('purchase', 'product', 10747), ('pro-
1812 duced_by', 'brand', 201), ('produced_by', 'product', 11772)] | path_prob_diff_user_mean=-
1813 0.4814049005508423 | path_entropy_diff_user_mean=0.02063143253326416 |
1814 path_rewards_diff_user_mean=4231.73046875
1815 **explainability_score: 1.8997045**
1816 **explainability_score: CI - Lower 1.6209097319169405**
1817 **explainability_score: CI - Upper 2.178499180741751 Count: 6**
1818 [ (11772, 'user 21001 has purchase product 10747 which was purchase by user 9748 who purchase
1819 product 11772',
1820 np.float32(0.8712647), np.float32(15.429942), (np.float32(3.6954694), np.float32(2.0),
1821 np.float32(1.0397208), np.float32(3274.7021),
1822 [('self_loop', 'user', 21001), ('purchase', 'product', 10747), ('purchase', 'user', 9748), ('purchase',
1823 'product', 11772)],
1824 np.float32(8.231993), np.float32(0.5185951), np.float32(-0.038260102), np.float32(1812.7737),
1825 np.float32(0.8712647), np.float64(0.5924699717088107), np.float64(1.150059420533621), 6)),
1826 (11772, 'user 21001 has purchase product 10747 which was also_viewed by related_product 23132
1827 who also_viewed product 11772',
1828 np.float32(1.4115256), np.float32(18.995838), (np.float32(3.6954694), np.float32(1.0),
1829 np.float32(1.0986123), np.float32(4418.364),
1830 [('self_loop', 'user', 21001), ('purchase', 'product', 10747), ('also_viewed', 'related_product', 23132),
1831 ('also_viewed', 'product', 11772)],
1832 np.float32(8.231993), np.float32(-0.4814049), np.float32(0.020631433), np.float32(2956.4353),
1833 np.float32(1.4115256), np.float64(1.1327308826966647), np.float64(1.690320331521475), 6)),
1834 (11772,**'user 21001 has purchase product 10747 which was produced_by by brand 201 who**
1835 **produced_by product 11772',**
1836 np.float32(1.8997045), np.float32(20.948555), (np.float32(3.6954694), np.float32(1.0),
1837 np.float32(1.0986123), np.float32(5693.659),
1838 [('self_loop', 'user', 21001), ('purchase', 'product', 10747), ('produced_by', 'brand', 201), ('pro-
1839 duced_by', 'product', 11772)],
1840 np.float32(8.231993), np.float32(-0.4814049), np.float32(0.020631433), np.float32(4231.7305),
1841 **np.float32(1.8997045), np.float64(1.6209097319169405), np.float64(2.178499180741751)**, 6)),
1842 (11772, 'user 21001 has purchase product 11808 which was also_viewed by related_product 23062
1843 who bought_together product 11772',
1844 np.float32(1.720593), np.float32(20.232107), (np.float32(3.6954694), np.float32(1.0),
1845 np.float32(1.0986123), np.float32(5225.7563),
1846 [('self_loop', 'user', 21001), ('purchase', 'product', 11808), ('also_viewed', 'related_product', 23062),
1847 ('bought_together', 'product', 11772)],
1848 np.float32(8.231993), np.float32(-0.4814049), np.float32(0.020631433), np.float32(3763.828),
1849 np.float32(1.720593), np.float64(1.44179825120405), np.float64(1.9993877000288602), 6)),
1850 (11772, 'user 21001 has purchase product 11808 which was also_viewed by related_product 23062
1851 who also_bought product 11772',
1852 np.float32(1.720593), np.float32(20.232107), (np.float32(3.6954694), np.float32(1.0),
1853 np.float32(1.0986123), np.float32(5225.7563),

[('self_loop', 'user', 21001), ('purchase', 'product', 11808), ('also_viewed', 'related_product', 23062),
('also_bought', 'product', 11772)],
np.float32(8.231993), np.float32(-0.4814049), np.float32(0.020631433), np.float32(3763.828),
np.float32(1.720593), np.float64(1.44179825120405), np.float64(1.9993877000288602), 6)),
(11772, 'user 21001 has purchase product 11808 which was also_viewed by related_product 23062
who also_viewed product 11772',
np.float32(1.720593), np.float32(20.232107), (np.float32(3.6954694), np.float32(1.0),
np.float32(1.0986123), np.float32(5225.7563),
[('self_loop', 'user', 21001), ('purchase', 'product', 11808), ('also_viewed', 'related_product', 23062),
('also_viewed', 'product', 11772)],
np.float32(8.231993), np.float32(-0.4814049), np.float32(0.020631433), np.float32(3763.828),
np.float32(1.720593), np.float64(1.44179825120405), np.float64(1.9993877000288602), 6))]

