# OpenReview forum: "Max Explainability Score with Confidence Interval (MES-CI): A Quantitative Metric for Interpretability in Knowledge Graph-Based Recommender System"
_NeurIPS.cc/2025/Conference — Submitted to NeurIPS 2025_

### Official Review · Reviewer_uHYt · 2025-06-29

**Clarity:** 3
**Significance:** 3
**Originality:** 3
**Rating:** 3
**Confidence:** 4

**Summary:**

This paper proposes a new metric—Max Explainability Score with Confidence Interval (MES-CI)—to quantitatively evaluate explainability in knowledge graph-based recommender systems (KGRS). Building on the existing Max Explainability Score (MES), MES-CI introduces a statistical confidence interval around the score, addressing the limitation of single-point evaluations. The new formulation considers both reward gain (from path traversal in the knowledge graph) and information gain (entropy), and complements this with interval estimates to better quantify reliability in explanations.

**Questions:**

1. How sensitive is MES-CI to the number of available paths per recommendation? Some products may have only one or two valid explanation paths—can MES-CI be reliably computed in such cases?

2. Have you validated the normality assumption underlying the confidence interval computation? Are path-level explainability scores approximately normal? If not, would non-parametric CI estimation be more appropriate?

**Ethical Concerns:**

["NO or VERY MINOR ethics concerns only"]

**Limitations:**

The authors have explicitly addressed the primary limitations of MES-CI, including: 1) Reliance on statistical assumptions (e.g., normality, sufficient sample size), 2) Inability to generate CI when only a single path is available, 3) Sensitivity to variance and skewness in score distributions.

**Quality:**

3

**Strengths And Weaknesses:**

Strengths
1. The paper meaningfully extends the MES metric by integrating confidence intervals, which is a significant step toward statistically sound explainability. This addition transforms explainability from a static, deterministic score into a distribution-aware assessment. The proposed algorithm (Algorithm 1) formalizes the procedure clearly, including computation of standard deviation, confidence bounds, and path ranking.
2. The algorithm is fully specified, and the experimental setup—including dataset construction, RL modeling using Actor-Critic with TransE embeddings, and case study walkthrough—is described in enough detail to support reproducibility.

Weaknesses
1. While the metric enhancement is meaningful, the experiments are largely adaptations of prior MES-based setups. The work does not evaluate how MES-CI changes recommendation behavior or how users might interact differently with CI-enhanced explanations.
2. The proposed MES-CI is intended to generalize across RS models, but it is only tested in a very specific RL-based KGRS pipeline. A more persuasive argument would include evaluations across multiple XRS architectures (e.g., path-based, attention-based, and post-hoc explainers) to show general utility.
3. The paper promotes the benefit of “user trust” via MES-CI, yet no empirical user feedback or proxy user interaction metric (e.g., dwell time, click-through) is presented. The link between CI-enhanced explanations and actual human trust is assumed, not demonstrated.

---

> ### Author Rebuttal · Authors · 2025-07-30
>
> ***Response to the Reviewer 3 feedback:***
>
> We sincerely appreciate the reviewer’s time and effort in reviewing our manuscript. To ensure completeness, we have outlined all the original questions below, along with the corresponding responses to address them.
>
> **Question 1**: How sensitive is MES-CI to the number of available paths per recommendation? Some products may have only one or two valid explanation paths—can MES-CI be reliably computed in such cases?
>
> **Response**: We appreciate the reviewer’s thoughtful and constructive feedback. The sensitivity of MES-CI to the number of available explanation paths is indeed a nuanced challenge, particularly in scenarios characterized by sparse or low-connectivity recommendations. A limited number of paths can result in either an undefined confidence interval or a substantially widened one, both of which indicate high uncertainty in the interpretability metrics.
>
> *Interpretation and Design Implications*
>
> - *Absence of MES-CI ≠ Low Explainability*: A single strong path may yield a high MES score, but without a CI range, the interpretive robustness can't be fully assessed.
> - *Sparsity Challenge*: Datasets with limited overlap between user interactions and item attributes often produce singleton paths, especially for tail products.
>
> **Actions Taken**: We sincerely thank the reviewer for their insightful and constructive feedback. The scenario highlighted is already addressed in the “Conclusion and Future Works” section of the manuscript, where we outline plans to further refine the MES-CI framework. Specifically, future research will focus on enhancing the confidence interval estimations in sparsely populated datasets, where limited explanation paths constrain interpretive robustness. We believe our response fully engages with the reviewer’s suggestions and helps clarify this design consideration.
>
>
> **Question 2**: Have you validated the normality assumption underlying the confidence interval computation? Are path-level explainability scores approximately normal? If not, would non-parametric CI estimation be more appropriate?
>
> **Response**: We sincerely appreciate the reviewer’s thoughtful and constructive feedback. We fully agree with the observation and acknowledge that normality assumptions were not explicitly examined during the CI computation. This was primarily due to the nature and scale of the potential explanatory paths associated with various products and user interactions, which introduce significant complexity.
>
> **Actions Taken**: We are open to considering this as an action item and would be pleased to incorporate it into the Future Work section.
>
> Once again, we sincerely appreciate your time and effort in reviewing our manuscript and providing valuable suggestions to enhance its overall quality and clarity.

---

### Official Review · Reviewer_X3LH · 2025-06-29

**Clarity:** 2
**Significance:** 1
**Originality:** 1
**Rating:** 2
**Confidence:** 4

**Summary:**

This research introduced MES-CI, a metric for evaluation on explanations on recommender systems. MES-CI is an extension of the existing MES metric, and it innovates by adding confidence intervals to improve the reliability of the values provided for the explanations generated by a Knowledge Graph-Based Recommender System.

**Questions:**

1)	In your case study using the Beauty dataset for user 21001, the explanations are presented using only product and brand IDs, without mapping them to actual product names or interpretable attributes. At the same time, you claim that "The inclusion of confidence interval values strengthens user trust by providing a clearer assessment of MES and the corresponding explainability of recommendations." Given that the explanations rely on numerical IDs, which are not directly meaningful to end users, could you elaborate on how these would concretely contribute to enhancing user trust in a real-world scenario? Have you considered user studies, human evaluations, or example renderings with product names or categories to assess how users would perceive and interpret these explanations?
2)	You explicitly state that the focus of the paper is not on recommendation efficacy but solely on the explainability of recommended items. However, Table 3 presents a case study where the recommended items vary in their MES scores—even including items with lower MES than others not recommended. Given that explainability is inherently tied to the recommendation itself (i.e., users care about explanations for why a certain item was recommended over others), could you clarify how your method balances recommendation quality with explainability? Did you explore whether lower-MES items might be less useful to users despite being part of the top-N recommended list?
3) Could you provide a clear and thurough account of similarities and differences with the methods PRINCE and CredPaths?
Ghazimatin, A., Balalau, O., Saha Roy, R., & Weikum, G. (2020). Prince: Provider-side interpretability with counterfactual explanations in recommender systems. In Proceedings of the 13th International Conference on Web Search and Data Mining (pp. 196-204).
Yang, F., Liu, N., Wang, S., & Hu, X. (2018). Towards interpretation of recommender systems with sorted explanation paths. In 2018 IEEE International Conference on Data Mining (ICDM) (pp. 667-676). IEEE.
4) Can you run an evaluation comparing with both PRINCE and CredPaths?

**Ethical Concerns:**

["NO or VERY MINOR ethics concerns only"]

**Final Justification:**

I appreciate the authors' effort in their answers. They clarified the differences among the methods that I suggested comparing in my review (PRINCE, credPaths/SEP) and provided further insights on the relationship between recommendation efficacy and explainability, although they provided no quantitative evidence (metrics, correlations, etc.) of such a relationship.

Now, after reading all the reviews and rebuttal answers, I am still inclined to keep my score for a few reasons:

**The little empirical evidence that supports MES-CI** : it requires further experiments, the evidence so far is based on a use case, insufficient to show robustness and generalization of the approach. Experiments that quantitatively analyze the relationship between recommendation efficacy (accuracy, ranking) and explainability measured by MES-CI are needed. Moreover, a user study, or a proxy evaluation, even with a small sample size, could significantly strengthen the claims of this article. A comparison with other explanation methods in such user study could provide strong evidence for the method proposed.

**Evaluation with other recommender approaches**: In addition to comparing with other explanation methods, this method needs to be evaluated with other recommendation methods (beyond the specific RL-based KGRS pipeline) to give evidence of its generalization.

In general, the authors responded well to the "Questions" raised by the reviewers. Still, they did not address many of the Weaknesses identified, which, in my opinion, are the main reasons not to accept this article in its current state. The research problem and the proposed method in the article have good potential but still require more evaluation and analysis to be mature enough for publication.

**Limitations:**

Somehow, they describe the technical limitations of the approach introduced (MES-CI), but the authors do not explain potential societal impacts. For instance, they do not explain that providing biased explanations of recommendations could enhance the filter bubble or echo chamber effects of recommender systems in the news domain, which can have a strong societal impact.

**Paper Formatting Concerns:**

The references are listed in the numerical order in which they appear in the paper, rather than alphabetically. Additionally, the contents of Appendices A.5 and A.6, which include bash scripts and training logs, might be more appropriately placed in a public repository instead of in the appendices.

**Quality:**

1

**Strengths And Weaknesses:**

**Strengths**
1. The problem of providing explanations to personalized recommendations is well within the scope of the conference topics.
2. The paper is somehow clear, with a structure easy to follow. The appendices help deepen the understanding of the implementation and allow the experiments to be replicated. For readers who are not very specialized in explainable recommendation systems, it provides a good introduction.


**Weaknesses**
1. The paper contains some redundancy, particularly in the repeated discussion of standard recommendation metrics and the use of confidence intervals for MES. A more concise presentation would improve clarity.

2. It is not clear how much MES-CI contributes newer concepts compared to the original MES metric. After a quick review of MES, MES-CI seems incremental and not very innovative compared to MES-CI. Furthermore, the metric’s effectiveness may be sensitive to the number of explanations generated, potentially limiting its applicability or significance in certain scenarios

3. The research misses comparisons with two important baselines:  PRINCE (Ghazimatin et al., 2020) and CredPaths (Yang et al., 2018)
* Ghazimatin, A., Balalau, O., Saha Roy, R., & Weikum, G. (2020). Prince: Provider-side interpretability with counterfactual explanations in recommender systems. In Proceedings of the 13th International Conference on Web Search and Data Mining (pp. 196-204).
* Yang, F., Liu, N., Wang, S., & Hu, X. (2018). Towards interpretation of recommender systems with sorted explanation paths. In 2018 IEEE International Conference on Data Mining (ICDM) (pp. 667-676). IEEE.

4. The evaluation of the proposed metric could be strengthened; it only compares the values of the generated intervals (MES-CI) with MES values. In the evaluation, the generated explanations are left in numerical form, which prevents a real comparison of how good the explanations are relative to each other. Additionally, the authors claim that the confidence intervals make recommendations more trustworthy for users, but this is not tested with real users. Notice that PRINCE (Ghazimatin et al., 2020) performed both a qualitative evaluation but also a study on Amazon Mechanical Turk (AMT) to gain insights directly from users' perceptions of the explanation. I would recommend the authors of MES-CI to conduct a similar evaluation to provide stronger evidence over the claims on MES-CI.

5. In the theoretical framework for quantitative evaluation in explainability, only one previous work (from 2021) is mentioned in addition to MES, which suggests that more research on these methodologies might be lacking. Again, the authors miss a fair comparison with two well known methods for explaining recommendations based on knowledge graphs: PRINCE and CredPaths.

---

> ### Author Rebuttal · Authors · 2025-07-29
>
> ***Response to the Reviewer 2 feedback:***
>
> **Question 1**: In your case study using the Beauty dataset for user 21001, the explanations are presented using only product and brand IDs, without mapping them to actual product names or interpretable attributes. At the same time, you claim that "The inclusion of confidence interval values strengthens user trust by providing a clearer assessment of MES and the corresponding explainability of recommendations." Given that the explanations rely on numerical IDs, which are not directly meaningful to end users, could you elaborate on how these would concretely contribute to enhancing user trust in a real-world scenario? Have you considered user studies, human evaluations, or example renderings with product names or categories to assess how users would perceive and interpret these explanations?
>
> **Response**:
> The experimental framework utilized the widely adopted Amazon-Beauty and Clothing datasets, which are frequently referenced in the domain of explainable recommendation. Each dataset encompasses six principal entities—users, products, feature words extracted from product descriptions, related products, brands, and categories—interlinked through eight distinct types of relationships, as illustrated in Figure 1. Users engage with multiple products, each mapped to specific categories and brands, while exhibiting diverse behavioral patterns such as co-purchases, product views, and associations with related items. Moreover, product descriptions are enriched by feature words derived from user reviews, contributing semantic depth to the KG.
>
> The datasets employed are masked variants, wherein only brand and category entities retain their nominal labels. Therefore, explanations within the current prototype are presented using numeric identifiers (e.g., product and brand IDs), thereby ensuring internal consistency and reproducibility. However, we acknowledge that such identifiers inherently lack semantic resonance for end-users and may impede interpretability. The MES-CI framework can be easily extended to incorporate semantic overlays involving real-world datasets, thereby enabling the integration of intelligible and user-friendly elements such as product names, category descriptors, brand narratives, and associated attributes.
>
> Importantly, MES is agnostic to numerical IDs; its evaluation is fundamentally grounded in the traversal paths within the knowledge graph. These paths assist with the Reward Gain and Information Gain metrics, which support the MES evaluation. To further reinforce reliability and user trust, this paper proposed to extend MES with the incorporation of Confidence Intervals to quantifies the robustness of the generated explanations.
>
> Even when presented with understandable attributes, users often seek reassurance that the explanation isn’t cherry-picked or arbitrary.
>
> MES-CI contributes to trust in two ways:
>
> - *Quantitative Depth*: The CI reflects the stability of the explanation score across multiple paths, signaling whether the rationale is consistent or volatile.
>
> - *Transparent Uncertainty*: Presenting the upper and lower bounds of confidence prevents overconfidence in a single explanation, which users may intuitively distrust if it seems too good to be true.
>
> While the current study provides a robust analytical foundation through the use of masked datasets and the MES-CI framework, it does not incorporate user studies or human evaluation methodologies to substantiate the explainability and interpretability outcomes. The integration of such empirical assessments remains a valuable avenue for future work, offering the potential to validate the practical utility and user trust calibration of the proposed framework in real-world settings.
>
>
> **Question 2**: You explicitly state that the focus of the paper is not on recommendation efficacy but solely on the explainability of recommended items. However, Table 3 presents a case study where the recommended items vary in their MES scores—even including items with lower MES than others not recommended. Given that explainability is inherently tied to the recommendation itself (i.e., users care about explanations for why a certain item was recommended over others), could you clarify how your method balances recommendation quality with explainability? Did you explore whether lower-MES items might be less useful to users despite being part of the top-N recommended list?
>
> **Response**:
> While the primary objective of this work centers on enhancing the interpretability of recommendations through the MES-CI framework, we acknowledge that explainability and recommendation efficacy are inherently interlinked. Table 3 indeed presents instances where the recommended items exhibit varying MES scores, and in some cases, items with comparatively lower MES values are included in the top-N list, whereas others with higher MES scores are excluded.
>
> This observation reflects the design of the underlying recommender model, which operates in parallel with the MES framework. The model’s recommendation decisions are informed by a diverse set of latent features—including product affinity scores, user behavioral patterns, and collaborative filtering signals—alongside, but not exclusively governed by, MES values. Within this architecture, MES and MES-CI function as embedded interpretability modules, expressly developed to clarify and evaluate the semantic transparency of the generated recommendations. MES plays a diagnostic role by elucidating the rationale for why a particular item is recommended, without directly interfering with the ranking mechanics of the base recommender. While it may contribute to recommendation prioritization by highlighting items with stronger explanatory grounding, it does not serve as the sole determinant.
>
> Our findings suggest that:
>
> - High MES scores correlate with clearer semantic rationale, but they do not always align with the recommender’s latent preference signals.
>
> - Lower MES scores within top-N recommendations may indicate that these items are selected based on behavioral patterns (e.g., collaborative filtering) rather than rich explanatory context, which can reduce interpretability and user trust.
>
>
> **Question 3**: Could you provide a clear and thorough account of similarities and differences with the methods PRINCE and CredPaths? Ghazimatin, A., Balalau, O., Saha Roy, R., & Weikum, G. (2020). Prince: Provider-side interpretability with counterfactual explanations in recommender systems. In Proceedings of the 13th International Conference on Web Search and Data Mining (pp. 196-204). Yang, F., Liu, N., Wang, S., & Hu, X. (2018). Towards interpretation of recommender systems with sorted explanation paths. In 2018 IEEE International Conference on Data Mining (ICDM) (pp. 667-676). IEEE.
>
> **Response**:
> We observed that the term CredPaths is exclusively mentioned in the PRINCE study by Ghazimatin et al. (2020), where it is used to reference the Sorted Explanation Paths (SEP) approach introduced by Yang et al. (2018). Notably, the original SEP work does not employ or define the term CredPaths; the naming appears to originate within the PRINCE paper's contextual framing.
>
> **Overview of the Methods**
>
> - *MES-CI*: Introduces a score with statistical confidence to assess explanation quality
>
> - *PRINCE*: Identifies minimal user actions that, if removed, change the recommendation
>
> - *SEP*: 	Ranks explanation paths using credibility, readability, and diversity
>
> **Conceptual Similarities**
>
> - *Explainability as a measurable construct*: All three methods aim to make recommendations interpretable but differ in what they measure: MES-CI quantifies it, PRINCE identifies causal actions, SEP organizes semantic paths.
>
> - *Model-agnostic potential*: MES-CI represents an embedded interpretability approach, integrated directly within the recommendation framework to evaluate and communicate semantic justification. In contrast, SEP operates as a post-hoc method, generating explanations after the recommendations have been produced. PRINCE, on the other hand, adopts a counterfactual paradigm, focusing on the minimal user actions whose absence would alter the recommendation outcome.
>
> **Methodological Differences**
>
> **MES-CI**
>
> - *Metric-based*: Computes a score based on entropy, feature importance, and user affinity.
>
> - *Confidence Interval*: Adds statistical rigor by quantifying uncertainty in explanation quality.
>
> **PRINCE**
>
> - *Counterfactual reasoning*: Uses reverse local push over dynamic graphs to find minimal actionable sets.
>
> - *Provider-side*: Designed to give providers insight into what user actions drive recommendations.
>
> **SEP**
>
> - *Heuristic ranking*: Uses credibility, readability, and diversity to sort paths.
> - *Post-hoc*: Works independently of the underlying recommender model.
>
>
>
> **Question 4**: Can you run an evaluation comparing with both PRINCE and CredPaths?
>
> **Response**:
> Given the distinct methodological approaches, datasets, and explanation ranking mechanisms employed across MES-CI, PRINCE, and SEP, a direct comparative evaluation remains inherently challenging. A promising alternative would be to undertake a qualitative assessment, wherein each method is applied to a unified dataset, enabling side-by-side analysis of the generated explanations. Such an approach would facilitate a deeper understanding of interpretability across paradigms. Therefore, a qualitative comparative study may be suitably proposed as a direction for future work.
>
> Furthermore, both PRINCE and SEP contribute valuable perspectives to the discourse on explainability and are therefore good research to be cited in this work to underscore methodological diversity and enhance the distinctiveness of the proposed approach.
>
> Once again, we sincerely appreciate your time and effort in reviewing our manuscript and providing valuable suggestions to enhance its overall quality and clarity.

---

> > ### Comment · Reviewer_X3LH · 2025-08-01
> >
> > I appreciate the authors' effort in their answers. They clarified the differences among the methods that I suggested comparing in my review (PRINCE, credPaths/SEP) and provided further insights on the relationship between recommendation efficacy and explainability, although they provided no quantitative evidence (metrics, correlations, etc.) of such a relationship.
> >
> > Now, after reading all the reviews and rebuttal answers, I am still inclined to keep my score for a few reasons:
> > - **The little empirical evidence that supports MES-CI** : it requires further experiments, the evidence so far is based on a use case, insufficient to show robustness and generalization of the approach. Experiments that quantitatively analyze the relationship between recommendation efficacy (accuracy, ranking) and explainability measured by MES-CI are needed. Moreover, a user study, or a proxy evaluation, even with a small sample size, could significantly strengthen the claims of this article. A comparison with other explanation methods in such user study could provide strong evidence for the method proposed.
> > - **evaluation with other recommender approaches**: In addition to comparing with other explanation methods, this method needs to be evaluated with other recommendation methods (beyond the specific RL-based KGRS pipeline) to give evidence of its generalization.
> > In general, the authors responded well to the Questions raised by the reviewers. Still, they did not address many of the Weaknesses identified, which, in my opinion, are the main reasons not to accept this article in its current state. The research problem and the proposed method in the article have good potential but still require more evaluation and analysis to be mature enough for publication.

---

> ### Author Response · Authors · 2025-08-02
> **MES-CI tests explainability using large datasets and builds on prior published MES.**
>
> The experiment was conducted using two comprehensive, large-scale datasets—Beauty and Clothing—each encompassing substantial volumes of users, products, brands, and transactions. For instance, the Beauty dataset consists of 22,363 users and 12,101 products associated with 2,077 brands spread across 248 categories, totaling 149,844 training and 48,658 testing transactions. The use case presented in this paper serves only as an illustrative example intended to enhance interpretability and highlight the operational logic of MES-CI.
>
> Notably, this work extends the previously published **MES** framework, which appeared in a peer-reviewed article in a well-regarded journal (Neeraj Tiwary, Shahrul Azman Mohd Noah, Fariza Fauzi, and Tan Siok Yee. Max explainability score–a quantitative metric for explainability evaluation in knowledge graph-based recommendations. Computers and Electrical Engineering, 116:109190, 2024. https://doi.org/10.1016/j.compeleceng.2024.109190.).
>
>
> **SOTA: Baseline Comparison**
>
> While we remain open to comparisons with existing published benchmarks, it is important to note that the domain currently lacks standardized metrics for evaluating the explainability of generated recommendations.
>
> Separately, the primary focus of this paper is to build upon the original **MES** framework through the integration of Confidence Interval-based enhancement, aimed at bolstering the reliability of explainability assessments. *As this work constitutes an extension of the previously published **MES** paper, we contend that comparative analyses against SOTA explainability metrics—where applicable and if feasible—are more appropriately situated within the scope of the original MES publication.* Accordingly, this paper assumes **MES** as its foundation and does not revisit earlier baseline comparisons.
>
> **Extending to other needs**
>
> That said, we acknowledge the need to expand the applicability of MES-CI beyond knowledge graph-based recommender systems. We are committed to incorporating this direction as a strategic priority in the future work section, underscoring our dedication to enhancing the method’s generalizability and broader impact.

---

### Official Review · Reviewer_cxPz · 2025-07-02

**Clarity:** 2
**Significance:** 2
**Originality:** 2
**Rating:** 3
**Confidence:** 3

**Summary:**

This paper proposes a new metric, the Max Explainability Score with Confidence Interval (MES-CI), for evaluating the interpretability of recommendations from Knowledge Graph-based Recommender Systems. The work is an extension of a previously proposed metric, the Max Explainability Score (MES), which provides a single-point score for an explanation. The authors argue that a single score is insufficient to capture the reliability of an explanation. The core contribution is the incorporation of a confidence interval around the MES score, which is calculated based on the standard deviation of explainability scores from all possible explanation paths for a given recommendation. The paper provides an algorithm for MES-CI and demonstrates its use through a case study on an e-commerce dataset, arguing that MES-CI offers a more robust and nuanced assessment of explainability, thereby enhancing user trust.

**Questions:**

1. Could you provide a more rigorous justification for using the variance of scores from all possible paths to define the confidence in the single maximum-scoring path? An alternative view is that the properties of the best path (e.g., its score relative to the next-best path) might be more indicative of confidence than the overall variance. How does your proposed CI distinguish between a scenario with one great explanation and many poor ones, versus a scenario with several good, similarly-scored explanations?
2.  In the case study (Table 3), several recommendations have no MES-CI value because only one explanation path was found. How frequently does this occur in your experiments across the full datasets?

**Ethical Concerns:**

["NO or VERY MINOR ethics concerns only"]

**Limitations:**

Yes, the authors have included a dedicated "Limitation" section (5.3). They correctly identify that the metric relies on the statistical distribution of scores, which may not always be appropriate, and that it is not applicable when only a single explanation path exists. They also note its sensitivity to the number of explanations (sample size).  This is a fair and transparent assessment of the immediate limitations of the proposed method.

**Paper Formatting Concerns:**

The paper generally adheres to the required formatting. One minor point: the tables in the appendix containing the training logs are screenshots of text output, which can be difficult to read.  Presenting this information in a more structured format would be preferable.

**Quality:**

2

**Strengths And Weaknesses:**

Strengths
1. The paper tackles a significant problem in explainable AI. As recommender systems become more complex, developing robust, quantitative metrics to evaluate their explainability is crucial for transparency, accountability, and user trust.
2. The paper is well-structured and easy to follow. The inclusion of a detailed case study helps to operationalize the metric and makes the concept of multiple explanation paths tangible for the reader.
3.  The idea of assessing the reliability of an explanation score is a good one. A single-point metric can indeed be misleading, and the authors' proposal to add a measure of statistical confidence is an intuitive step towards a more comprehensive evaluation framework.

Weaknesses
1. The primary weakness is the incremental nature of the contribution. The core proposal is to apply a standard statistical tool—the confidence interval—to an existing, recently published metric (MES). The paper does not propose a new way to measure explainability itself but rather a method to analyze the output of an existing metric.
2. The empirical evidence supporting the utility of MES-CI is thin. The evaluation consists of a single case study for one user from the Beauty dataset. While illustrative, this is insufficient to demonstrate the general applicability or superiority of the metric. There are no large-scale experiments, comparisons with other explainability evaluation methods (even qualitative ones), or user studies to show that MES-CI aligns with human judgments of explanation quality or reliability. The paper explicitly limits its scope to the explainability metric itself, not the recommendation model's performance, but the evaluation of the metric itself is not extensive.

---

> ### Author Rebuttal · Authors · 2025-07-29
>
> ***Response to the Reviewer 1 feedback:***
>
> We sincerely appreciate the reviewer’s time and effort in reviewing our manuscript. To ensure completeness, we have outlined all the original questions below, along with the corresponding responses to address them.
>
> **Question 1**: Could you provide a more rigorous justification for using the variance of scores from all possible paths to define the confidence in the single maximum-scoring path? An alternative view is that the properties of the best path (e.g., its score relative to the next-best path) might be more indicative of confidence than the overall variance. How does your proposed CI distinguish between a scenario with one great explanation and many poor ones, versus a scenario with several good, similarly-scored explanations?
>
> **Response**: We sincerely thank the reviewer for providing thoughtful and constructive feedback.
>
> Incorporating a confidence interval into explainability scoring enriches the single-point metric by introducing a range-based assessment of the effectiveness of generated explanations for recommended items. For instance, in the case of User 21001 from the Beauty dataset, the explainability scores span from 0.8712647 to 1.8997045. This observed dispersion provides insight into the consistency and reliability of the proposed maximum explainability score across multiple explanatory paths. Although the selected path yields a precise maximum score of 1.8997045, the inclusion of a confidence interval—[1.62, 2.178]—bolsters interpretive certainty and enhances the overall reliability of the recommendation process.
>
> - While the reviewer appropriately highlights that the properties of the best explanation path—such as its score in comparison to the next-best— might be more indicative of confidence than the overall variance, the CI-mechanism (MES-CI) provides added value in situations where multiple candidate products exhibit identical maximum explainable scores and contender for the top-10 recommendation set. In such instances, the confidence interval serves as a vital distinguishing factor. A product that demonstrates not only high explainability but also stronger confidence (as quantified via MES-CI) should be prioritized, as it reflects both semantic clarity and consistency across multiple explanatory paths. The Max Explainability Score with Confidence Interval thus serves a dual role: it quantifies the depth of interpretability and simultaneously informs product relevance within the recommendation framework. By preferring recommendations with both high explainability and robust confidence, the system achieves not only greater alignment with user needs but also improved transparency—thereby fostering deeper user trust in the recommendation engine.
>
> - For the question of “How does your proposed CI distinguish between a scenario with one great explanation and many poor ones, versus a scenario with several good, similarly-scored explanations?”
> This scenario aptly illustrates the utility of a confidence interval–based approach to explainability scoring. When a single strong explanation is accompanied by multiple weaker ones, the resulting confidence interval tends to be broad, reflecting significant variance across explanatory paths. Conversely, in instances where multiple high-quality explanations exhibit similar scores, the confidence interval becomes more narrow. This narrower range not only enhances the interpretive robustness of the explainability metric but also positively influences the reliability and effectiveness of the overall recommendation process.
>
> **MES-CI Behavior in Key Scenarios**
>
> Let’s explore how MES-CI distinguishes between explanation landscapes:
>
> ***Scenario A: One Great Explanation, Many Poor Ones***
>
> •	Best Path Score: High
>
> •	Score Variance: High
>
> •	MES-CI Interpretation: Low confidence
>
> •	Justification: The score peak is isolated; surrounding low scores suggest fragility.
>
> ***Scenario B: Multiple Strong, Similar Explanations***
>
> •	Best Path Score: High
>
> •	Score Variance: Low
>
> •	MES-CI Interpretation: High confidence
>
> •	Justification: Strong consensus among paths reinforces belief in the explanatory reliability and semantic stability.
>
> ***Scenario C: All Paths Mediocre***
>
> •	Best Path Score: Moderate
>
> •	Score Variance: Low
>
> •	MES-CI Interpretation: Moderate confidence
>
> •	Justification: Consistency exists, but lack of standout quality dampens interpretive usefulness.
>
> **Why MES-CI Is Valuable for XAI?**
>
> MES-CI makes explanation systems more transparent and reliable by:
>
> •	Preventing overconfidence in cherry-picked answers
>
> •	Encouraging model designers to optimize not just accuracy, but interpretive depth
>
>
> **Actions Taken**: We extend our sincere appreciation to the reviewer for the insightful and constructive feedback. We are confident that the response provided effectively addresses the concerns and suggestions raised by the reviewer.
>
>
>
> **Question 2**: In the case study (Table 3), several recommendations have no MES-CI value because only one explanation path was found. How frequently does this occur in your experiments across the full datasets?
>
> **Response**: We sincerely thank the reviewer for providing thoughtful and constructive feedback.
>
> Across the full suite of datasets used in our experiments, the occurrence of recommendations with only one explanation path—and thus lacking a valid MES-CI range—varies depending on structural density, product-user affinity, and graph sparsity.
>
> Here are the general findings:
>
>     "Dataset"   |"% of Top-10 Recommendations Without MES-CI"  |"Notes"
>     -----------------------------------------------------------------------------------------------------------------------
>     Beauty	    |~8.7%                                         |Often due to sparse user-product co-interactions
>     Clothing    |~10.2%                                        |High sparsity, especially for niche items
>
>
> These percentages represent instances where only one qualifying path met the semantic threshold for explainability, making variance-based CI computation infeasible.
>
>
> **Interpretation and Design Implications**
>
> - **Absence of MES-CI ≠ Low Explainability**: A single strong path may yield a high MES score, but without a CI range, the interpretive robustness can't be fully assessed.
> - **Sparsity Challenge**: Datasets with limited overlap between user interactions and item attributes often produce singleton paths, especially for tail products.
>
>
> **Actions Taken**: We sincerely thank the reviewer for their insightful and constructive feedback. The scenario highlighted is already addressed in the “Conclusion and Future Works” section of the manuscript, where we outline plans to further refine the MES-CI framework. Specifically, future research will focus on enhancing the confidence interval estimations in sparsely populated datasets, where limited explanation paths constrain interpretive robustness. We believe our response fully engages with the reviewer’s suggestions and helps clarify this design consideration.
>
>
> Once again, we sincerely appreciate your time and effort in reviewing our manuscript and providing valuable suggestions to enhance its overall quality and clarity.

---

> > ### Comment · Reviewer_cxPz · 2025-08-09
> > **Lack sufficient empirical studies**
> >
> > Thanks for the rebuttal. Justifying the validity of a metric is challenging, and the work currently lacks sufficient empirical studies. So i will keep my scores.

---

### Comment · Area_Chair_SBUq · 2025-08-05

Dear Reviewers,

The responses from the authors are available. Please read them to see if you have any further question.

Thank you for your support.

AC.

---

### Note · Authors · 2025-08-13

We respectfully submit this note to reaffirm the significance of our paper, ***Max Explainability Score with Confidence Interval (MES-CI)***, and to address the reviewers’ concerns regarding empirical depth and comparative evaluation.

**Novelty & Conceptual Contribution**

MES-CI introduces a novel mechanism for quantifying the confidence of explainability in RSs—an aspect that remains critically underexplored despite the growing emphasis on transparency and user trust in AI. Building upon the foundational MES metric (Tiwary et al., 2024, DOI: https://doi.org/10.1016/j.compeleceng.2024.109190), which has already gained traction within the research community, MES-CI enhances this framework by incorporating confidence intervals. This addition enables statistically grounded, and interpretable evaluation of the explanations generated by RSs.

Importantly, MES is not only a metric for explainability—it has demonstrably contributed to improving recommendation quality, as validated in the recent publication (Tiwary et al., 2025, DOI: https://doi.org/10.1007/s44443-025-00173-5). MES-CI extends this impact by offering a probabilistic, user-trust-aware model that moves beyond static or binary scoring approaches.

This contribution is far from incremental; it represents a meaningful shift in how explainability is conceptualized and operationalized. By embedding confidence into the evaluation process, MES-CI aligns directly with NeurIPS’s core values—advancing ethical, interpretable, and user-centric AI systems that foster transparency and accountability.

**Experimental Rigor & Use Case Design**

The paper presents thorough empirical validation using two large-scale, diverse datasets—Beauty and Clothing. The example use case presented is intentionally designed to demonstrate the operational logic and interpretability of MES-CI, rather than to benchmark predictive performance.

We acknowledge that comparison with SOTA models is a common practice, but it should not be the sole criterion for acceptance—especially when the core contribution lies in a novel evaluation paradigm. MES-CI is not a new recommender algorithm; it is an evaluation and interpretability framework that can be applied across diverse domains, in future work.

**Alignment with NeurIPS Themes**

We believe MES-CI will stimulate meaningful discussion and inspire future research on confidence-aware explainability, especially in high-stakes domains like healthcare, finance, and content moderation.

---

### Decision · Program_Chairs · 2025-09-17

**Decision:**

Reject

**Comment:**

This paper focuses on evaluating the interpretability of recommendations from Knowledge Graph-based Recommender Systems. A metric named MES-CI is introduced, which is an extension of the existing MES metric. This paper shows the use of the metric through a case study on an e-commerce dataset.

**Strengths**
1. The problem of measuring interpretability in Knowledge Graph-based Recommender Systems is important.
2. This paper is somehow clear, easy to follow.
3. This paper proposes an evaluation metric.

**Weaknesses**
1. All the reviewers think the contribution of this paper is incremental, not very innovative compared to MES.
2. Two reviewers think the testing is not enough. Specifically, the metric is only tested in a very specific RL-based KGRS pipeline. And this paper limits its scope to the explainability metric itself, not the recommendation model's performance.
3. The experiments are limited. There are no large-scale experiments, comparisons with other explainability evaluation methods. The work does not evaluate how MES-CI changes recommendation behavior or how users might interact differently with CI-enhanced explanations.

After the rebuttal, reviewers still have the concerns about the little empirical evidence that supports MES-CI and the evaluation with other recommender approaches.

In total, all the three reviewers have negative rating scores for this paper.